# Coupling the regional climate model ICON-CLM v2.6.6 into the Earth system model GCOAST-AHOI v2.0 using OASIS3-MCT v4.0

Ha Thi Minh Ho-Hagemann[1], Vera Maurer[2], Stefan Poll[3], Irina Fast[4]

[1]Institute of Coastal Research, Helmholtz-Zentrum Hereon, Geesthacht, Germany (Ha.Hagemann@hereon.de)

[2]Deutscher Wetterdienst, Offenbach, Germany

[3] Institute of Bio and Geosciences Agrosphere (IBG-3), Forschungszentrum Jülich, Jülich, Germany; CASA-SDL Terrestrial Systems, Jülich Supercomputing Centre (JSC), Jülich, Germany

[4]German Climate Computing Center (DKRZ), Hamburg, Germany

Correspondence to: Ha Thi Minh Ho-Hagemann, Ha.Hagemann@hereon.de

## Abstract

Interactions and feedback between compartments of the Earth system can have a significant impact on local and regional climate and its changes due to global warming. These effects can be better represented by regional Earth system models (RESMs) than by traditional stand-alone atmosphere and ocean models. Here, we present the RESM GCOAST-AHOI version 2.0, which includes a new atmospheric component, the regional climate model ICON-CLM, which is coupled with the ocean model NEMO and the hydrological discharge model HD via the OASIS3-MCT coupler. The GCOAST-AHOI model has been developed and applied for climate simulations over the EURO-CORDEX domain. Two 11-year simulations from 2008-2018 of the uncoupled ICON-CLM and GCOAST-AHOI give similar results for seasonal and annual means of near-surface air temperature, precipitation, mean sea level pressure and wind speed at 10 m height. However, GCOAST-AHOI has a cold SST bias of 1-2 K over the Baltic and the North Seas, most pronounced in winter and spring seasons. A possible reason for the cold SST bias could be the underestimation of the downward shortwave radiation at the surface of ICON-CLM with the current model settings. Despite the cold SST bias, GCOAST-AHOI was able to capture other key variables such as those mentioned above well. Therefore, GCOAST-AHOI can be a useful tool for long-term climate simulations over the EURO-CORDEX domain. Compared with the stand-alone NEMO3.6 forced by ERA5 and ORAS5 boundary forcing, GCOAST-AHOI has positive biases in sea ice fraction and salinity, but negative biases in runoff, which need to be further investigated in the future to improve the coupled simulations. The new OASIS3-MCT coupling interface OMCI implemented in the ICON-CLM model adds a possibility to couple ICON-CLM with an external ocean model and an external hydrological discharge model using OASIS3-MCT instead of the YAC coupler. Using OMCI, it is also possible to set up a RESM with ICON-CLM and other ocean and hydrology models possessing the OASIS3-MCT interface for other regions, such as the Mediterranean Sea.

Keyword: GCOAST, ICON-CLM, OASIS3-MCT coupling interface, climate simulations, EURO-CORDEX, RESM

# 1   Introduction

GCOAST (Geesthacht Coupled cOAstal model SysTem) is an Earth system framework developed at Helmholtz-Zentrum Hereon, Germany (Staneva et al., 2018). GCOAST is a modular system of

different models each developed for a specific component of the Earth system. Based on a specific
scientific question, different models from GCOAST can be selected. These models can be plugged
together by various couplers, such as OASIS3-MCT (Craig et al., 2017), ESMF (Earth System Modeling
Framework; Hill et al., 2004), or FABM (Framework for Aquatic Biogeochemical Models;
https://fabm.net). The coupling can be done at different levels of coupling granularity and the
couplers handle the exchange of information between model combinations, individual models, and
processes.
GCOAST systems have been applied for several studies covering the Baltic and-North Sea region
and part of the North Atlantic. These studies include atmosphere-river-ocean-sea ice coupling (Ho-
Hagemann et al., 2020), atmosphere-wave coupling (Wahle et al., 2017; Wiese et al., 2019, 2020),
wave-ocean coupling (Staneva et al., 2016; Schloen et al., 2017; Lewis et al., 2019), hydrosphere-
biosphere coupling for the Elbe estuary (Pein et al., 2019), the total organic carbon-macrobenthos
coupling model (Zhang et al., 2019), and multi-model couplings developed by Lemmen et al. (2018),
which have been applied to assess ecosystem impacts of offshore wind farms (Slavik et al., 2019).
So far, the atmospheric model component of GCOAST has been the non-hydrostatic limited area
atmospheric model COSMO-CLM v5.0 (Rockel et al., 2008). The COSMO (cOnsortium for Small-scale
mOdeling) model was initially developed by the Deutscher Wetterdienst (DWD, the German
Meteorological Service) in the 2000s as a limited-area weather forecast model. Later, it was further
developed in the Climate Limited-area Modeling Community (CLM-Community) as the regional
climate model COSMO-CLM (hereafter referred to as CCLM). In December 2021, COSMO v6.0 was
released which is the last version of the COSMO model. With this release, the development of the
COSMO model ended after more than two decades. The successor of the COSMO model is the ICON
model.
In 2001, a cooperation between the DWD and the Max-Planck Institute for Meteorology (MPI-M)
was initiated, with the aim of developing a new modelling system for weather prediction and climate
simulations. As one result of this initiative, the global numerical weather prediction model ICON
(Icosahedral Nonhydrostatic) was developed (Zängl et al., 2015). Nowadays, with contributions from
the Karlsruhe Institute of Technology (KIT) and the German Climate Computing Center (DKRZ), etc.,
the ICON Earth system framework can include not only the atmospheric, land, river routing, ocean-
sea ice, wave, and biogeochemical compartments but also the Aerosols and Reactive Trace gases
(ART) model. ICON can be set up to operate on several high-performance computing systems such
as Bull ATOS at DKRZ (Hamburg, Germany), NEC-Aurora Tsubasa at DWD (Offenbach, Germany) or
BullSequana at Forschungszentrum Jülich (FZJ, Jülich, Germany). ICON can be used on a wide range

of scales from climate projection, climate prediction and numerical weather prediction down to large-eddy simulations (Dipankar et al., 2015; Heinze et al., 2017).

The atmospheric component of ICON includes two different physics packages: the first one is the Numerical Weather Physics package of DWD (i.e. the ICON-NWP model); and the second one is the ECHAM physics package of MPI-M (i.e. the ICON-A model, Giorgetta et al., 2018). The global atmospheric model ICON-A is coupled with the global ocean model ICON-O (Korn, 2017) and the land and biosphere model JSBACH (Reick et al., 2021) within the ICON Earth System Model (ICON-ESM; Jungclaus et al., 2022). ICON-NWP can be also coupled with ICON-O in the ICON-Seamless Earth system coupling framework which has been newly developed during the recent years. In ICON-Seamless, there are two options for the land surface schemes, TERRA and JSBACH, which are coupled via subroutine to the atmospheric component. A new land surface model (ICON-Land) is being developed based on JSBACH and some features of TERRA. Another component of ICON-Seamless is the wave model ICON-Wave. The hydrological discharge model HD can now be used as an external model instead of being coupled as a subroutine of JSBACH.

The components of ICON are coupled together using YAC (Yet Another Coupler; Hanke et al., 2016). However, coupling between ICON components, or the coupling of multiple ICON components to an external model without a YAC coupling interface is not supported due to how the initial communicator splitting is implemented.

ICON can also be used in a configuration with regional grid refinement (2-way nesting) or in limited area mode. ICON-LAM is the Limited-Area Mode of ICON-NWP. Starting in 2017, DWD and the CLM-Community decided to develop the climate limited area mode (ICON-CLM, Pham et al., 2021) based on ICON-LAM. Within ICON-Seamless, a limited area mode of the ocean model (ICON-O-LAM) is being developed and can be coupled with ICON-LAM via YAC.

As mentioned above, coupling a component or multiple components of ICON to an external model that has no YAC coupling interface is not supported and potentially impossible. For the coupling of ICON-CLM as an atmosphere component into GCOAST-AHOI, which includes HD and the ocean model NEMO (Nucleus for European Modelling of the Ocean, Madec et al. 2017), representing the ocean and sea ice components, basically, there were two feasible options: either to implement a YAC interface in NEMO and HD, or to implement an OASIS interface in ICON-CLM. For the option one, the YAC coupling interface was added into the HD source code by M. Hanke (DKRZ) (see Hagemann et al. 2023), but YAC has not yet been available in the NEMO source code. To our current knowledge, there is no RESM with NEMO using the YAC coupler. The NEMO model is already linked to the OASIS coupler, which has been used to couple NEMO with many other model components.

Implementing the YAC interface in NEMO would require a larger effort, as the NEMO source code is
much more complicated than the HD code. In addition, although the NEMO source code is freely
available, we are ordinary users in the NEMO community, not members of the model development
team. Therefore, implementing and especially maintaining the YAC interfaces in NEMO is a big
challenge. For the second option, HD and NEMO already have the OASIS3-MCT coupling interface
(OMCI), so all we had to do was to implement OMCI in ICON. Here, we also have the advantage that
we belong to the ICON development team of the CLM-Community, so that we can get great and
quick technical support from the development team when coding with ICON. Therefore, in 2021,
we started to port the OASIS coupling interfaces from CCLM to ICON-CLM for the coupling to NEMO
and HD.
Some other groups were using a similar method while coupling ICON into their available coupled
system model which does not include the YAC coupler. For example, Bauer et al. (2021) have
implemented the ESMF interfaces in an earlier version of ICON-NWP as well as into the ocean model
GETM to build up the regional ocean-atmosphere coupling over the Baltic Sea. However, they did
not consider sea ice in the coupling. There is an ongoing work at FZJ to couple ICON-CLM with the
Community Land Model (CLM) via the OASIS3-MCT coupler (manuscript in preparation) as has been
done for the CCLM via the OASIS interface (see Shrestha et al., 2014; Will et al., 2017).
The aim of this article is to give a detailed description of the OASIS3-MCT coupling interface
(hereafter referred to as OMCI) in ICON-CLM (ICON release version 2.6.6), how to implement OMCI
with as little modification of the ICON source code as possible, how to compile it on the high-
performance computing system Levante at DKRZ, and how to run the coupled system model
GCOAST-AHOI with ICON-CLM for climate simulations over the EURO-CORDEX domain. This
information is useful to other groups planning to couple ICON-CLM with NEMO or any other ocean
model that already has an OASIS3-MCT interface available. The Earth System Modelling (ESM)
Community agrees that ICON and IFS (coupled to FESOM and NEMO) will play a central role within
the Helmholtz Association of German Research Centres (HGF). This new OMCI opens more
opportunities to use ICON-CLM in ESM applications as well as in other modelling communities. The
OMCI can also be applied to couple with a land surface model with minor necessary adaptations.
We briefly introduce the coupled system model GCOAST-AHOI in Section 2 and describe the
details of OMCI in ICON-CLM in Section 3. Experiment setups are presented in Section 4, followed
by an analysis of the model simulations in Section 5. Finally, conclusions and a discussion are given
in Section 6.

## 2 The coupled system model GCOAST-AHOI

GCOAST-AHOI is a subset of GCOAST that includes model components for **A**-Atmosphere and Land, **H**-Hydrological discharge, **O**-Ocean, and **I**-Sea Ice. GCOAST-AHOI version 1.0 (Ho-Hagemann et al., 2020) contains the atmospheric model CCLM v5.0, the ocean model NEMO v3.6 (including the sea ice model LIM3) and the hydrological discharge model HD v4.0 (Hagemann and Dümenil, 1998; Hagemann et al., 2020), coupled via OASIS3-MCT v2.0. A detailed description of CCLM, NEMO and HD as components of GCOAST-AHOI can be found in Ho-Hagemann et al. (2020).

In the GCOAST-AHOI version 2.0, ICON-CLM replaces CCLM as atmospheric model, which is coupled to NEMO v3.6 and HD v5.1 via OASIS3-MCT v4.0. By coupling the atmosphere-ocean-river runoff models in GCOAST-AHOI, we aim to close the water balance in the RESM. Figure 1 illustrates the three models exchanging radiation, wind, pressure, temperature, humidity, water, and sea ice related variables at their interfaces via the OASIS coupler.

The OMCI in NEMO v3.6 has been modified compared to the original one in the officially released version at http://forge.ipsl.jussieu.fr/nemo/wiki/Users/release-3.6 to be able to receive state variables from the atmospheric model (Ho-Hagemann, 2024). Supplement S1 contains a flowchart of the OMCI for NEMO v3.6. This flowchart differs slightly from Figure 9 in Will et al. (2017), who used the older version NEMO v3.3. The OMCI in HD can be found in the source code publication of Hagemann and Ho-Hagemann (2021) and Hagemann et al. (2023). Supplement S2 shows the OMCI of HD. In this article, we describe in detail the OMCI in ICON-CLM.

In Section 3, we demonstrate the construction of the OMCI in ICON-CLM and the optional coupling methods between ICON-CLM and NEMO.

## 3 The coupling OASIS3-MCT interface in ICON

### 3.1 Interface structure

Figure 2 shows a flowchart of ICON with the OMCI implemented for coupling with NEMO and HD. 10 levels of ICON's source code are described: the first level is the main program ICON, the second level starts with the *start_mpi*, then *atmo_model* and ends with *stop_mpi*, etc.

Levels 2 to 6, 8 and 9 comprise subroutines of ICON (marked in red) that are modified by the coupling. On levels 3 to 7 and 10, new subroutines (orange boxes B1-B7) have been added with the OMCI. They are organized in three modules (cpl_oas_vardef.f90, cpl_oas_mpi.f90 and cpl_oas_interface.f90) containing about 3000 lines of Fortran code (including the current debug lines). The files have been added to the icon/externals/oasis3-mct directory and linked to the src/atm_phy_nwp directory of the ICON source tree. The detailed description of the interface

structure can be found in Supplementary S4.
Supplement S5 contains a guide for compiling ICON with this OMCI on Levante at DKRZ. The
preparation of OASIS input files for GCOAST-AHOI is described in Supplement S6, which is
accompanied by an example of the namcouple file in Supplement S7 and the namelist_cpl_atm_oce
in Supplement S8. The command to run GCOAST-AHOI on Levante is provided in Supplement S9.
The complete package to conduct experiments for this study is included in the Starter Package for
ICON-CLM Experiments (SPICE; Rockel and Geyer, 2022), which is a workflow engine to easily
perform long-term simulations. This tool has been further developed from the ICON-CLM_SP starter
package (Pham et al. 2021). Some additional parts for coupling with NEMO and HD have been added
to the original package.

### *3.2   Coupling methods*

In the officially released version of NEMO v3.6, several fluxes and variables, including shortwave
(SW) and longwave (LW) radiation fluxes, latent (LH) and sensible heat (SH) fluxes, rain, snow,
evaporation, ice sublimation, mean sea level pressure (MSLP) and surface momentum, can be sent
from an atmospheric model to NEMO via the OASIS3-MCT coupler. To be able to receive state
variables from the atmospheric model, the OMCI in NEMO v3.6 has been modified to allow air
temperature and air specific humidity at 2 m height (T_2M and QV_2M respectively) to be sent from
the atmospheric model to NEMO. This allows NEMO to use these variables to calculate the LH and
SH, as in the case of the stand-alone NEMO using the "CORE bulk formulae" (Large and Yeager,
2004). Thus, we have three options for the coupling method between ICON and NEMO:
a) CPL_flx: **flux coupling**, which is the default option in the NEMO source code (described above)
b) CPL_var: **state variable coupling**, the new method, where SW and LW, T_2M, QV_2M, wind

speed at 10 m height (UV_10M), rain, snow, MSLP and surface momentum are sent from

ICON-CLM to NEMO. NEMO calculates LH and SH using the "CORE bulk formulae" which is

based on the Monin Obukhov similarity theory.

c) CPL_mix: **mixture coupling**, the new method, like CPL_var, but ICON-CLM also sends LH and

SH to NEMO. NEMO then averages them with the LH and SH calculated using the "CORE bulk

formulae".

With the modification of OMCI in NEMO v3.6, it is now easy to select the coupling method via
the namelist settings. Section 5 considers the simulations using the coupling method 3 (CPL_mix),
which was also used in Ho-Hagemann et al. (2020). An extra experiment using the coupling method
1 (CPL_flx) is also conducted and analyzed in Section 5.
In turn, NEMO sends the sea surface temperature, sea ice fraction and sea ice albedo to ICON-
CLM. Figure 3 illustrates how the surface temperature is updated in ICON over the ocean (left side)
and over land (right side) in the presence of sea ice and snow. ICON utilizes a tile approach to
compute surface fluxes of momentum and scalars. For the "sea-water type" grid boxes, the grid box
mean fluxes are computed as a weighted average of the fluxes over ice and over open water, using
the fractional ice cover *fice* and the fractional open water cover (*1 − fice*) as the respective weights.
Sea ice in each ICON grid box is considered only if *fice* exceeds its minimum value of 0.015.
Otherwise, the grid box is treated as ice-free. In ICON, two types of surface temperature are
considered: the ground temperature t_g and the surface temperature t_s. If a grid box is covered
by sea ice or snow, t_g is the mixed temperature of the free sea ice/free snow surface temperature
and the temperature on top of the sea ice/snow. Under the sea ice, t_s is calculated as a mixture of
the free sea ice temperature and the salt water freezing temperature of 271.45 K. If there is no sea
ice or snow in the grid box, t_g is equal to t_s. In principle, NEMO can send the mixed sea ice and
water temperature to ICON to update t_g over the ocean points, as in CCLM in Ho-Hagemann et al.
(2020). Or it can send the open water temperature, the sea ice surface temperature and the sea ice
fraction so that ICON can calculate t_g as the mixture. However, in the uncoupled mode of the
current ICON-CLM version, the sea surface temperature (SST) forcing is read in as the variable
t_seasfc (or t_s_w in Fig. 3) and passes through the subroutines *nwp_surface_init* and
*process_sst_and_seaice* to calculate t_g. To be consistent with the ICON-CLM updates, we pass the
SST (to update the t_seasfc), the sea ice fraction (to update fr_seaice), and the sea ice albedo
(alb_si_ext) from NEMO to ICON. ICON will then calculate t_g, t_s, alb_si, etc. using its sea ice
scheme. In the future, we may modify this coupling method by using the sea ice temperature from
NEMO.
**4   Experimental design**
In this study, four main experiments are conducted for the period of 2008-2018 (Table 2). The
uncoupled ICON (**ICON266**), the coupled GCOAST-AHOI (**ICPL266**), the stand-alone NEMO v3.6
(**NEMO3.6**), and the stand-alone HD v5.1 forced by ICON266 runoffs (**HDICON266**). Two additional
experiments are conducted: **ICPL266_noNewa** as a sensitivity test for the location of the Newa River
mouth on NEMO grid, and **ICPL266_flx** to test the coupling method CPL_flx.
Each experiment starts on 01 January 2008 and ends on 01 January 2019, restarting each month.
The integration domains of ICON, NEMO and HD are displayed in Fig. 4. The namelist setup of
physical parametrization for ICON-CLM is similar to that of the NUKLEUS project (B. Geyer, personal
communication). The resolution of ICON is R13B5, with an approximate mesh size of 12 km, using
60 vertical levels. The model top height is at 23.5 km. The following physical schemes are used in
the current namelist setting of ICON: Radiation scheme ecRad (Hogan and Bozzo, 2018; Rieger et
al., 2019); Mass-flux shallow and deep convection scheme (Tiedtke, 1989; Bechtold et al., 2008);
Microphysics single-moment scheme (Doms et al., 2004); Planetary boundary layer scheme
prognostic TKE (Raschendorfer, 2001; Raupach and Shaw, 1982); Land-surface scheme tiled TERRA
(Schrodin and Heise, 2001; Schulz et al., 2016; Schulz and Vogel, 2020). The initial and lateral
boundary forcing of ICON is obtained from the ERA5 reanalysis data (Hersbach et al., 2020). The
Tegen aerosol climatology (Tegen, 1997), i.e. a monthly aerosol optical depth of sulphate droplets,
total dust, organic carbon, black carbon, and sea salt, is used in this study. The initial and daily lateral
boundary forcing of NEMO is taken from the ORAS5 reanalysis data (Copernicus Climate Change
Service, 2021). The spatial resolution of NEMO is ~3.7 km with 50 vertical levels. In the stand-alone
mode, NEMO3.6 is driven by the atmospheric ERA5 data, the ocean ORAS5 data and climatological
river runoff data. HD has the resolution of 1/12 degrees, ca. 8 km. More information on the model
configuration can be found in Table 1 and in Ho-Hagemann et al. (2020).
To estimate the computational performance of the coupled model, we used LUCIA (Maisonnave
and Caubel, 2014), which is part of OASIS3-MCT. In Supplement S10 and Fig. S1, we describe how to
use LUCIA for GCOAST-AHOI to optimize the computational performance.

## 5  Evaluation of model simulations

The first two years 2008-2009 are excluded as spin-up time, and the output data of the two
simulations ICON266 and ICPL266 for nine years (2010-2018) are compared with the observational
and the ERA5 reanalysis data to assess the model performance. For sea surface temperature (SST),
we use the Operational Sea Surface Temperature and Ice Analysis (OSTIA) data (Good et al. 2020)
to evaluate the simulated SST of ICPL266. For air temperature at 2 m height (T_2M) and precipitation
(TOT_PREC), the daily E-OBS data (Haylock et al. 2008; Van den Besselaar et al. 2011) version 27.0
on the grid of 0.11 degree are used. The ERA5 reanalysis data are interpolated onto the E-OBS grid
and used as a reference for comparison with the simulated shortwave and longwave surface
radiation, turbulent fluxes, mean sea level pressure (PMSL), wind speed at 10 m height (SP_10M),
and T_2M. The Surface Radiation Data Set - Heliosat (SARAH) - Edition 2 (Pfeifroth et al. 2017) is
used to evaluate the shortwave downward radiation of the simulations.
Seasonal means of winter (DJF), spring (MAM), summer (JJA), autumn (SON) and annual means
(ANN) of several variables are analyzed in this section.

## 5.1 Sea surface temperature and 2m air temperature

Over the ocean, the sea surface temperature (SST) of ICON266 is the ERA5 forcing data, which is based on observations, so it's very close to the OSTIA data (not shown). Thus, the SST difference between the coupled and the stand-alone ICON-CLM simulations (Fig. S2) can be interpreted as a bias towards a measurement-based product. In the coupled model, the SST is provided by NEMO over the GCOAST domain. In general, ICPL266 has a cold SST bias of about 1-3 °C compared to OSTIA over the GCOAST domain, except around the British coast in summer (JJA, Fig. 5a). The SST bias of ICPL266_flx (Fig. 5b) is similar to ICPL266 (Fig. 5a). Ho-Hagemann et al. (2020) noted that using CPL_flx when coupling COSMO-CLM with NEMO leads to larger biases in the SST than using CPL_mix. This is not the case here when coupling ICON-CLM. A possible reason for this is that due to the tile approach (cf. Sect. 3.2), the fluxes from ICON-CLM to NEMO are sent separately over water and sea ice, while COSMO-CLM v5.0 doesn't have the tile approach, therefore, the fluxes in each ocean grid box sent from the atmosphere to the ocean are the mixed fluxes of water and sea ice.

The annual mean SST bias of the stand-alone NEMO3.6 is less than 0.5 °C over the Baltic and North Seas, and of about -1 to -2 °C over the North Atlantic compared to the OSTIA data (Fig. 5c). In summer, a positive SST bias of about 1-2 °C is present over the Baltic and North Seas. In this case, the reduction of SST by the coupling reduces the warm bias of the stand-alone NEMO3.6.

The cold SST bias of ICPL266 over the GCOAST domain intensifies the cold T_2M bias (Fig. 6b, Fig. S2b), especially in winter (DJF) and spring (MAM). In summer, ICPL266 reduces the warm T_2M bias of ICON266 (Fig. 6a). In general, the annual (ANN) T_2M bias of ICPL266 is slightly more negative by about 0.5 °C than that of ICON266. ICPL266_flx reproduces a similar T_2M (Fig. 6c) to that of ICPL266. Comparison with the E-OBS data (Fig. S3) shows similar results to Fig. 6, except over Northern Africa and Turkey, where the quality of the E-OBS data is affected by the lack of observations in that region (cf. Fig. 1 in Hagemann and Stacke, 2022).

## 5.2 Shortwave radiation, longwave radiation, and turbulent fluxes

A possible reason for the SST cold bias of ICPL266 may be that the shortwave and longwave radiation from ICON-CLM sent to NEMO is too low. Figure S4 shows the relative bias (%) of the shortwave downward radiation (SWDN) of ICON266 and ICPL266 compared to the ERA5 data, as well as the relative difference (%) between SARAH2 and ERA5. Figure 7 shows a zoomed section of Fig. S4 over the GCOAST ocean domain (note the adapted color scale). In general, both ICON266 and ICPL266 have a positive SWDN bias of less than 10 % over land compared to ERA5, except for the larger bias of 15-20 % over northern Europe in winter and eastern Europe in autumn (Fig. S4). Over the North Sea, ICON266 and ICPL266 have a small negative bias of about 5-10 % compared to ERA5 (Fig. 7).

The area of negative SWDN bias in the North Sea is slightly larger in ICPL266 than in ICON266.
Comparing the ERA5 data and the SARAH2 data, SWDN over southern Europe is similar between the
two datasets, with SARAH2 being slightly larger over land (Fig. S4). In general, the SWDN of ICON266
and ICPL266 over the North Sea is rather close to the SARAH2 data, but slightly overestimated over
the Baltic Sea.
Figure S5 in the Supplementary Appendix shows a similar plot to Fig. S4, but for the longwave
downward radiation (LWDN) and without the SARAH2 data, as it is not available. The modeled
LWDN has a negative bias of about 2-4 % annually and a larger bias in winter of about 6-8 %, most
pronounced over land. Over the ocean, ICON266 reproduces well the LWDN of the ERA5 data, and
ICPL266 has a small negative bias of 2-4 %. Overall, the negative SWDN bias over the North Sea and
the negative LWDN bias of ICON-CLM give an indication why the ICPL266 shows an increased cold
SST bias.
The ERA5 reanalysis data is used as the atmospheric forcing for the uncoupled NEMO3.6, and the
namelist settings of the NEMO model used in this study were tuned for an SST close to OSTIA (Fig.
5c). If the same namelist settings of NEMO3.6 are used for ICPL266, to reduce the cold SST bias over
the North Sea in the coupled simulations, a bias correction for SWDN and LWDN should be done.
Figures S6a, b and Figures S7 a, b show the seasonal SWDN and LWDN of ICPL266 and NEMO3.6
averaged over the North Sea and Baltic Sea for the period of 2010-2018. Note that we don't show
the result of ICPL266_flx in Figs. S6 and S7 because there is no output of LWDN in ICPL266_flx due
to the setup of the CPL_flx coupling method. Over the North Sea, the SWDN of ICPL266 is smaller
than the ERA5 used for NEMO3.6 in spring and summer (Fig. S6a), which mainly leads to the cold
SST bias of ICPL266 (Fig. 5a). Therefore, we plan to increase the SWDN of ICON by about 10 % before
sending it to NEMO. However, the cold SST bias over the Baltic Sea does not seem to be directly
related to the SWDN as there is no clear SWDN difference between ICPL266 and NEMO3.6 in
summer or in any other season (Fig. S6a). The LWDN of ICPL266 is similar to NEMO3.6 in summer
but slightly smaller in the other three seasons, both over the North Sea and the Baltic Sea. Increasing
the LWDN in ICON-CLM by about 5-10 W/m2 before sending it to NEMO should be tested to reduce
the SST bias. Note that the seasonal cycle of LWDN is more pronounced over the North Sea than
over the Baltic Sea.
We also compare the turbulent heat flux (i.e. the sum of SH and LH) of NEMO3.6 and the flux of
NEMO in ICPL266, averaged over the North Sea and the Baltic Sea (Fig. S6c and Fig. S7c), and the
net downward heat flux, which is the sum of SWDN, LWDN, SH and LH (Fig. S6d and Fig. S7d). We
only consider the turbulent flux because NEMO doesn't write out SH and LH separately, but only the
SWDN, LWDN and the net downward heat flux. Note that the turbulent flux from NEMO3.6 is
calculated using the "CORE bulk formulae", and due to the CPLmix coupling method used, the
turbulent flux in NEMO from ICPL266 is the average of the flux from ICON-CLM and the one
calculated inside NEMO (see section 3.2). The results are similar for both seas. The turbulent flux
and the net downward flux of the two experiments are quite similar with the largest differences in
winter (DJF) and summer (JJA).

Using ERA5 as a reference, the SH and LH biases of ICON266 and ICPL266 are shown in Fig. S8

and Fig. S9. Over land, the bias of ICON266 is very similar to that of ICPL266. However, over the
ocean, the bias of ICPL266 is generally more positive (i.e. the fluxes are less negative) than that of
ICON266 with the largest bias over the North Atlantic. Smaller heat fluxes are consistent with lower
SSTs in ICPL266, as lower SSTs lead to larger stability and less vertical mixing. Over the North and
Baltic Seas, SH and LH of ICPL266 are quite close to ERA5. Despite of the SST forcing from ERA5,
ICON266 has a negative SH bias of about -5 to -15 W/m$^2$ over the North and Baltic Seas, especially
in winter. This suggests a future analysis of the difference in air temperature at the lowest level of
ICON-CLM and ERA5.

Besides the energy flux biases causing the cold SST bias, a spin-up time of 2 years may be too

short for NEMO to reach the stable state, leading to the cold SST bias. In addition, NEMO's namelist
settings should also be optimized for the coupled simulations.

Currently, in the COPAT2 (Coordinated Parameter Testing, phase 2) initiative of the CLM-

Community, several parameters of ICON-CLM are being tested in a similar way as done for the
COSMO-CLM model (Russo et al., 2024) to find out the recommended settings. For example, the use
of the transient aerosol MAC2-SP (Kinne, 2019; Stevens et al., 2017) and a careful adjustment of
various namelist settings related to cloud cover, the soil and vegetation scheme and the turbulent
transfer will further reduce the T_2M cold bias and improve the shortwave downward radiation.
### *5.3  Precipitation, mean sea level pressure, and wind speed*
The precipitation biases (Fig. 9) of the three simulations ICON266, ICPL266 and ICPL266_flx
compared to the E-OBS data are very similar in general, with a wet bias in winter and spring and a
dry bias in summer (JJA) and autumn (SON). Fig. S10 and Fig. S11 in the Supplementary show the
biases of PMSL and SP_10M of ICON266 and ICPL266 compared to ERA5. The PMSL and SP_10M
figures of ICPL266_flx are not shown because they are very similar to those of ICPL266.

ICPL266 tends to overestimate the PMSL throughout the year except in the summer, while

ICON266 has only a pronounced positive bias in winter (DJF) and negative bias in summer (JJA). The
wind speed of the two simulations is very similar over land (Fig. S11). ICPL266 reduces the wind
speed over the GCOAST ocean domain by up to 1.5 m s$^{-1}$ compared to ICON266 (Figs. 8, S11).
Therefore, while ICON266 has a positive bias of about 0.5 m s$^{-1}$ over the North Sea and the Baltic
Sea in winter (Fig. S11a), ICPL266 is very close to ERA5 (Fig. S11b). In general, ICPL266 produces a
cooler SST and weaker wind speed than ICON266, consistent with the smaller SH mentioned above
in Section 5.2. This positive feedback is known as the thermal feedback (TFB) mechanism in the
atmosphere-ocean surface coupling process (Zhang and Perrie, 2001; Renault et al., 2023).
Figure 10 shows monthly climatology of different variables (T_S, T_2M, TOT_PREC, PMSL) over
the GCOAST domain or the whole EURO-CORDEX domain, considering only ocean or land points.
ICPL266 has a cold T_S bias of about 1-2 °C over the ocean (Fig. 10a), which also causes the T_2M
bias of 0.5-1 °C over the ocean (Fig. 10c). In winter, ICPL266 is slightly colder over land than ICON266
and E-OBS (Fig. 10d). In summer, both simulations are very close to E-OBS. The simulated
precipitation of ICON266 tends to be overestimated compared to E-OBS with a maximum in May
and June, and slightly underestimated in August and September (Fig. 10b). The coupled run shows
1-3 mm/month less precipitation than the atmosphere-only experiment. In previous studies by Ho-
Hagemann et al. (2015, 2017), the stand-alone atmospheric model COSMO-CLM has a dry bias in
summer and the coupled run reduces the dry bias due to the improvement of the moisture
convergence and transport from ocean to land. This situation is not found in the current study,
which needs to be thoroughly analyzed in the future.
For the PMSL, the whole EURO-CORDEX domain is considered, but separately for ocean points
(Fig. 10e) and land points (Fig. 10f). In both cases, ICPL266 has a larger PMSL than ICON266. The
higher surface pressure in ICPL266 may be caused by the cooler air near the surface (due to the
negative T_2M bias) which leads to a higher density of the air mass and therefore a higher pressure.
Over the ocean, the PMSL of ERA5 is better reproduced by ICPL266 than by ICON266. Over land,
ICPL266 increases the PMSL positive bias in winter compared to ICON266. ICPL266_flx and ICPL266
have similar results (not shown) indicating that the coupling methods used in GCOAST-AHOI v2.0
don't affect the simulated climate variables strongly in this study.
*5.4   Sea ice*
The sea ice fraction bias of ICPL266 is about 0.2-0.3 over the Bothnian Bay and Sea in winter and
spring (Fig. 11a), while ICPL266_flx has a larger bias of about 0.3-0.5 (Fig. 11b) and the ERA5-forced
NEMO3.6 has a relatively small positive ice fraction bias there (Fig. 11c). The monthly mean sea ice
fraction averaged over the Bothnian Bay and Sea from ICPL266 and NEMO3.6 compared to the
OSTIA data is shown in Fig. 11d, where the sea ice temporal variation is quite well captured by the
two models, with a high peak in spring 2010 and a low peak in spring 2015. However, all three
simulations overestimate the sea ice fraction of OSTIA, with the two coupled simulations showing a
larger positive bias also in the time series. While ICPL266 has a winter SWDN about 8 % larger than
ERA5 (Fig. 7b), the incoming shortwave radiation is relatively small over the high latitudes in winter.
Therefore, we don't expect the positive SWDN bias to be the main reason for the overestimation of
sea ice. However, the LWDN of ICPL266 is about 10 W/m$^2$ lower than ERA5 (the forcing for
NEMO3.6) in winter over the North and Baltic Seas (Figs. S6b, S7b), and T_2M is about 3 $^o$C lower
than ERA5 over the Scandinavian region surrounding the Bothnian Bay and Sea (Fig. 6b). The cold
T_2M bias and negative LWDN bias of ICPL266 may explain its positive sea ice fraction bias. The cold
air temperature above the sea ice surface often produces more sea ice in winter and spring,
especially over an area with water of low salinity, such as the Bothnian Bay and Sea. Figure 6c shows
the larger T_2M cold bias of ICPL266_flx over the Baltic Sea in spring about 1$^o$C more than ICPL266
(Fig. 6b) which is consistent with the larger sea ice fraction bias of ICPL266_flx (Fig. 11b) compared
with that of ICPL266 (Fig. 11a).

Another factor that could contribute to an increase in sea ice cover in spring would be an increase

in river runoff, which would result in less salty sea water and therefore more sea ice. These two
variables are analyzed in the next section.
### *5.5   Salinity and river runoff*
As mentioned in section 4, NEMO3.6 uses a climatological dataset for river runoff. Therefore, a
rough verification of the river runoff produced by the HD model in ICPL266 can be made by
comparing against this climatological river runoff. Differences in sea surface salinity and river runoff
between ICPL266 and NEMO3.6 are shown in Figure 12. The salinity simulated by ICPL266 is about
0.3-1 PSU higher than that of NEMO3.6 along the British Isles and the North Sea coast, and about
0.9-1.8 PSU higher in the Baltic Sea. Two areas with the largest salinity differences of more than 2
PSU are found south of the Kattegat and in the Gulf of Finland (Fig. 12a). The river runoff differences
(Fig. 12b) are largest near the Ems and Newa estuaries with more than 0.1 and 0.2 kg/m2/s,
respectively. The small river runoff difference between the two models at Kattegat cannot be used
to directly explain the increase in salinity there. The river runoff differences near the Ems estuary
have opposite signs (blue point overlaid by a red one in Fig. 12b), but very similar values. The reason
for this may be the discrepancy in the locations of river mouths between the NEMO3.6 setup, where
the river runoff is taken from a climatology, and those in ICPL266, which are defined based on the
river mouths in the HD model and the NEMO land-sea mask. In the latter case, the river mouths of
the HD model are interpolated onto the NEMO grid by searching for the closest ocean point of
NEMO. For example, the Ems River mouth in ICPL266 may not be in the same position as in the
climatology data. This discrepancy would lead to a difference in salinity near the coast (see Fig. 12a).
The extent of the effect on salinity in the deeper layers of the ocean in a longer-term simulation
needs to be analyzed in the future.

The large difference in river runoff near the Newa estuary is also caused by a mismatch in the

locations of the river mouths. In this case, the mouth of the Newa River in the climatology data
(60.1333°N, 29.888°E) is located slightly northwest of its "real" location (Wikipedia: 59.9453°N,
30.1708°E). The interpolation program used to define the mouths of the HD rivers on the NEMO
grid, by searching for the closest ocean point to the mouth of the HD rivers, found the mouth of the
Newa River at (59.95835°N, 30.20825°E), which is very close to the "real" position and at the
furthest grid point to the east in the Gulf of Finland. However, in the NEMO model, this eastern
boundary point in the Gulf of Finland is masked as a buffer zone. Therefore, the discharge from HD
to NEMO at this point in ICPL266_noNewa was ignored in the NEMO calculations, resulting in a lack
of freshwater inflow to the Gulf of Finland in ICPL266 (Fig. 12b2) and consequently an increase in
salinity (Fig. 12a2). The ICPL266 simulation with the Newa River mouth located on the NEMO buffer
zone is referred to ICPL266_noNewa in Figure 12. To overcome the location deficiency, that the
Newa River mouth was shifted one grid point to the west on the NEMO grid to allow the large
amount of river runoff to enter the Gulf of Finland in the coupled model. Therefore, the salinity
difference of ICPL266 compared to NEMO3.6 is reduced (Fig. 12a1) and the river runoff difference
shows the shift of the river mouth instead of the missing one (Fig. 12b1). The shift of the Newa River
mouth has little effect on the simulated atmospheric variables, but improves the simulated salinity
in the Baltic Sea, which is important for ecosystem modelling when a marine biogeochemical or
ecosystem model such as ECOSMO (Daewel and Schrum, 2013) is coupled with GCOAST-AHOI in the
future.

Other river mouths in the Baltic Sea have river runoff differences of less than 1.4 kg/m$^2$/s when

comparing ICPL266 to NEMO3.6. In general, ICPL266 tends to simulate less river runoff than the
climatology, leading to increased salinity there. The sources of the river runoff used for NEMO in
ICPL266 are the surface and sub-surface runoffs from the land component in ICON-CLM that are
transported to the ocean by HD. We applied the HD model to calculate the discharge using the
ICPL266 and ICON266 surface and subsurface runoff (Table 3) to evaluate it against the discharge
observation. The annual discharge difference of ICPL266 and HDICON266 in the Baltic Sea is about
-11 %. However, HDICON266 with a discharge of 12449 m$^3$/s is about -20 % biased towards the
HELCOM (Helsinki Commission; Svendsen and Gustafsson, 2022) value of 15676 m$^3$/s. Note that for
Baltic Sea ocean models, the mean long-term bias of river runoff must be less than 7 % (Hagemann
and Stacke, 2022). In the North Sea, ICPL266 discharge is about -4 % compared to HDICON266, which
has an annual value of 6366 m$^3$/s. However, both models have a dry discharge bias compared to the
OSPAR data (Farkas and Skarbøvik, 2021), which is 9190 m$^3$/s.
The main driver of the runoff is precipitation. Figure 9 shows that over Scandinavia, ICON266 has
a wet bias of about 10-30 mm/month in spring and summer compared to the E-OBS data. Thus, even
with the wet precipitation bias, ICON266 has a dry discharge bias. At the same time, ICPL266
precipitation is lower than for ICON266 in spring and summer, and therefore closer to the E-OBS
data (Fig. 9b). This difference in precipitation between ICPL266 and ICON266 explains the -11 %
difference in discharge, which increases the dry discharge bias. The reduction of the precipitation
bias in ICPL266 leading to a larger discharge dry bias implies that a better simulation of precipitation
compared to observations does not necessarily lead to a better runoff. We note that the runoff from
the atmosphere-only ICON-CLM has a general dry bias, which can be attributed to the respective
parametrizations in the TERRA land surface scheme used in ICON-CLM (S. Hagemann, pers.
communication, 2024). This dry discharge bias can be improved either by using the JSBACH land-
surface model in ICON-Seamless, or by applying a discharge bias correction developed by Hagemann
et al. (2024).
In Section 5.4, it was speculated that an increase in river runoff would lead to less saline sea
water and therefore more sea ice over the Baltic Sea. In our study, ICPL266 simulates too little river
runoff, leading to increased salinity in the Baltic Sea, which would mean less sea ice. However, the
sea ice fraction is increased compared to the ERA5-forced NEMO3.6. Thus, the main factor causing
the bias in the sea ice fraction seems to be the cold bias in the air temperature over sea ice in the
Baltic Sea in winter and spring (Fig. 6b and c).

## Conclusion and Outlook

In the present study, we introduce the regional Earth system model (RESM) GCOAST-AHOI
version 2.0, in which a new atmospheric component - the regional climate model ICON-CLM version
2.6.6 - is coupled with the ocean model NEMO version 3.6 and the hydrological discharge model HD
version 5.1 via the OASIS3-MCT coupler version 4.0.
GCOAST-AHOI v2.0 is developed and applied for climate simulations over the EURO-CORDEX
domain. Several 11-year simulations from 2008-2018 of the uncoupled ICON-CLM (ICON266) and
GCOAST-AHOI (ICPL266, ICPL266_flx, ICPL266_noNewa) yield similar results for seasonal and annual
means of near-surface air temperature and precipitation, as well as mean sea level pressure and
wind speed at 10 m height. However, GCOAST-AHOI has a cold SST bias of 1-2 °C over the Baltic and
the North Seas, most pronounced in winter and spring seasons. The coupling methods CPL_mix and
CPL_flx give similar biases of SST and other climate variables like T_2M, precipitation, PMSL, etc.
A possible reason for the cold SST bias of GCOAST-AHOI could be the underestimation of the
downward shortwave radiation at the surface of ICON-CLM with the current model settings. A
deeper analysis of the bias will be done in the next study, especially after re-running the simulations
with the optimal settings of ICON-CLM, which will be found within the COPAT2 initiative of the CLM-
Community. For example, the performance of ICON-CLM will be tuned by using the transient
MACv2-SP aerosol data (Kinne, 2019) and modified namelist parameters related to cloud cover to
improve the shortwave downward radiation and reduce the cold bias.
Despite the cold SST bias, GCOAST-AHOI was able to capture the distribution of temperature,
precipitation, mean sea level pressure and wind speed well, similar to the uncoupled ICON-CLM
model. However, GCOAST-AHOI provides larger biases in sea ice fraction and salinity over the Baltic
Sea compared to the stand-alone ocean simulation (NEMO3.6) forced by ERA5 and ORAS5. The sea
ice fraction bias is related to the cold T_2M bias in ICPL266 and ICPL266_flx. Using the flux coupling
method CPL_flx instead of CPL_mix doesn't strongly affect the bias of SST and climate variables but
causes a larger sea ice fraction positive bias over the Baltic Sea. In the future study, a new simulation
of ICPL266 with the CPL_var coupling method will be conducted and compared with the current
ICPL266 and ICPL266_flx experiments to investigate the impact of the coupling methods on the sea
ice simulation.
The salinity bias is attributed to the dry runoff bias of ICPL266 compared to the climatology, with
the largest bias values are found near the Ems and Newa estuaries. The dry runoff bias near the Ems
and Newa River mouths is due to a mismatch of the river mouth locations between the climatology
and ICPL266. An adjustment of the Newa River mouth location must be made to allow the Newa
River runoff to flow into the Gulf of Finland. The effect of the river runoff bias on salinity in the
deeper layers of the ocean should be analyzed in the future.
In addition, added value of the coupled model compared to the stand-alone model is usually
found in the case of extreme events (Ho-Hagemann et al., 2015, 2017, 2020; Wiese et al., 2019,
2020). Therefore, we will analyze the model simulations with a focus on extreme events in the next
study.
Our present study shows that the RESM GCOAST-AHOI can be a useful tool for conducting long-
term regional climate simulations. The new OASIS3-MCT coupling interface OMCI implemented in
the ICON-CLM model adds a possibility to couple ICON-CLM with an external ocean model and an
external hydrological discharge model, not only with NEMO and HD, using OASIS3-MCT instead of
YAC. Given that the standalone model components for each the atmosphere and the ocean are
available for a specific geographical domain, it is also quite easy to apply GCOAST-AHOI to other
regions. Besides preparing the lateral boundary conditions for NEMO over the new domain, and the
OASIS input files (as described in Supplementary S6 and S7), it is necessary to prepare several new
parameter files so that OASIS3-MCT can exchange the discharge from HD to NEMO without
interpolation. On the one hand, these are files for the general setup of the HD model. The creation
of these files is described in Sect. 3 of the HD model readme mark down file included in the HD
model package (Hagemann et al. 2023). On the other hand, this includes the HD model coupling file,
which is used for coupling via OASIS. Instructions for its generation are provided in Section 2.1 of a
markdown file dedicated to the HD model coupling exercises (Hagemann et al. 2023).

ICON-CLM with OMCI is also used to couple ICON-CLM with NEMO v4.2 over the GCOAST domain

(manuscript in preparation) and with NEMO-MED v3.6 over the Mediterranean Sea region in the
CLM-Community. OMCI for the older ICON version 2.6.4 can be found in Ho-Hagemann (2022).

Recently, the ICON Consortium has developed and released the Community Interface (ComIn)

for the ICON model to allow ICON to be coupled with external model components. The main
challenge for the external model component coupling is the initial splitting of MPI_COMM_WORLD,
which is done in ICON by a grouping of the mpi communicators (MPI-handshake) (M. Hanke, pers.
communication, 2024). There are about 40 ComIn entry points in the new release version of ICON.
Using the ComIn entry points does not require any additional patching of the ICON source code. A
coupling interface to an external model such as OMCI would have to be moved into a ComIn-plugin
to connect to the entry points in the ICON source code. In addition, the communicator splitting using
the MPI-handshake algorithm would have to be implemented in the NEMO and HD source code.

Currently, also a limited area mode of the ocean model (ICON-O-LAM) is being developed within

the ICON consortium. This can be coupled with ICON-CLM via the YAC coupler in the ICON-Seamless
framework. When that RESM will be available in the future and will be applied for the EURO-CORDEX
domain, its simulation can be compared with the simulations of GCOAST-AHOI as a good reference.
Investigating difference in simulations of the two RESMs could be helpful to understand better the
coupling interactions and feedback between model components of the climate system.

**Supplementary Materials**: Supplementary material is available online together with the submitted manuscript.
**Author Contributions:** H.T.M. H.-H. developed the OMCI in ICON-CLM and HD, modified the OMCI in NEMO, designed

the experiments and carried them out, analyzed the results; H.T.M. H.-H. prepared the manuscript with contributions from all co-authors; V.M. contributed to analyze the simulations; S.P contributed to develop the OMCI in ICON-CLM; I.F. supported debugging the GCOAST-AHOI on the DKRZ HPC system. All authors have read and agreed to the published version of the manuscript.

**Funding**: This study was conducted within the CoastalFutures project that was funded by the German Federal Ministry of Education and Research under grant number 03F0911E. Moreover, it has been supported by funding from the German project REKLIM. The work described in this article has received funding from the Initiative and Networking Fund of the Helmholtz Association through the project "Advanced Earth System Modelling Capacity (ESM)". The content of the article is the sole responsibility of the author(s) and it does not represent the opinion of the Helmholtz Association, and the Helmholtz Association is not responsible for any use that might be made of the information contained. The study also contributes to the fourth programm-oriented funding phase (PoF IV) of the Helmholtz Association of German Research Centers.

**Acknowledgements**: The authors are grateful to the following entities: The German Climate Computing Center (DKRZ) provided the computer hardware for the Limited Area Modelling simulations in the project "Regional Atmospheric Modelling"; We acknowledge the E-OBS dataset from the EU-FP6 project UERRA (http://www.uerra.eu) and the Copernicus Climate Change Service, and the data providers in the ECA18ndD project (https://www.ecad.eu). We appreciate the use of the ERA5 reanalysis product that was provided by the European Centre for Medium-Range Weather Forecasts (ECMWF). ERA5 data reformatted by the CLM community provided via the DKRZ data pool were used. This study has been conducted using E.U. Copernicus Marine Service Information (https://doi.org/10.48670/moi-00165). We express our thanks to CERFACS (France) for the availability of the OASIS3-MCT coupler, especially to Eric Maisonnave for a support with the LUCIA tool in OASIS3-MCT. We thank Sebastian Grayek (formerly at Helmholtz-Zentrum Hereon) for preparing the NEMO lateral boundary conditions. We are also grateful Daniel Rieger and Daniel Reinert (DWD) for their advice on ICON source code. We thank Stefan Hagemann (Hereon) for proving the information on setting the HD model, and thank Beate Geyer (Hereon) for the information on NUKLEUS settings for the ICON-CLM model. We express our thanks to Panagiotis Adamidis (DKRZ) for the great technical support, and thanks to Moritz Hanke (DKRZ) for his comments and related information on YAC and ComIn in ICON. We acknowledge the valuable comments of the topical editor Sophie Valcke (CERFACS) and two anonymous reviewers.

**Code and data availability:** ICON is available to the community under a permissive open source licence (BSD-3C). One can download the newest released version at https://gitlab.dkrz.de/icon/icon-model. The source code of ICON v2.6.6 including the OMCI is published on Zenodo (https://doi.org/10.5281/zenodo.11057794). Note that ICON v2.6.6 was released before the open source release of ICON. As this version still comprises 3rd party modules with a more restrictive license, we had to change the file access from public to available upon request.

The NEMO source code is freely available and distributed under CeCILL license (GNU GPL compatible). To download the NEMO reference version (for now revision 3.6):

*svn co http://forge.ipsl.jussieu.fr/nemo/svn/NEMO/releases/release-3.6/NEMOGCM*

The modified NEMO v3.6 source code for different coupling methods are published on Zenodo (https://doi.org/10.5281/zenodo.11057794).

The HD source code is available at https://doi.org/10.5281/zenodo.4893099.

Source code of OASIS3-MCT v4.0 with small modifications in lib/psmile/src/GPTLget_memusage.c and
lib/mct/mct/m_AttrVectComms.F90 is published on Zenodo (https://doi.org/10.5281/zenodo.11057794).
Input data, run-scripts, evaluation scripts are published on Zenodo (https://doi.org/10.5281/zenodo.11057794).
Because of its huge volume, forcing data used for this study is available from the authors upon request.
**Conflicts of Interest**: The authors declare no conflict of interest. The funders had no role in the design of the study; in
the collection, analyses, or interpretation of data; in the writing of the manuscript, or in the decision to publish the
results.

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

*Table 1: Model configuration*

| Configuration | ICON | NEMO | HD | Coupler OASIS3-MCT |
|---|---|---|---|---|
| **Version** | v2.6.6 | v3.6 | v5.1 | v4.0 |
| **Domain** | EURO-CORDEX | North Sea, Baltic Sea, North Atlantic | Europe | - |
| **Resolution** | ~ 12 km | ~ 3.7 km | ~ 8 km | - |
| **Grid points** | 231660 | 902 x 777 | 960 x 540 | |
| **Time step** | 100 s | 90 s | 3600 s | 3600 s |
| **Forcing** | ERA5 | ORAS5, OTIS | - | - |


*Table 2: Model experiments*

| Experiment | Description |
|---|---|
| **ICON266** | Uncoupled ICON-CLM v2.6.6 forced by ERA5 |
| **ICPL266** | Coupled GCOAST-AHOI forced by ERA5 and ORAS5 |
| **ICPL266_noNewa** | Coupled GCOAST-AHOI forced by ERA5 and ORAS5, Newa River mouth is located on the buffer zone of NEMO grid. |
| **ICPL266_flx** | Coupled GCOAST-AHOI forced by ERA5 and ORAS5, coupling method CPL_flx |
| **NEMO3.6** | Stand-alone NEMO v3.6 forced by ERA5 and ORAS5 |
| **HDICON266** | Stand-alone HD v5.1 forced by surface runoff and sub-surface runoff of ICON266 |


*Table 3: Seasonal discharge ($m^3/s$) of ICPL266 and HDICON266 summed over the North Sea, the Baltic Sea during the period of 2010-2018. Diff. (%) is the difference of ICPL266 to HDICON266.*

| Areas | North Sea | | | Baltic Sea | | |
|---|---|---|---|---|---|---|
| Seasons | ICPL266 | HDICON266 | Diff. (%) | ICPL266 | HDICON266 | Diff. (%) |
| DJF | 7356 | 7687 | -3.31 | 9148 | 10864 | -15.80 |
| MAM | 6896 | 7438 | -5.42 | 18788 | 19884 | -5.51 |
| JJA | 5103 | 5482 | -3.79 | 9837 | 10990 | -10.49 |
| SON | 4352 | 4857 | -5.05 | 6755 | 8056 | -16.15 |
| ANN | 5927 | 6366 | -4.39 | 11132 | 12449 | -10.58 |



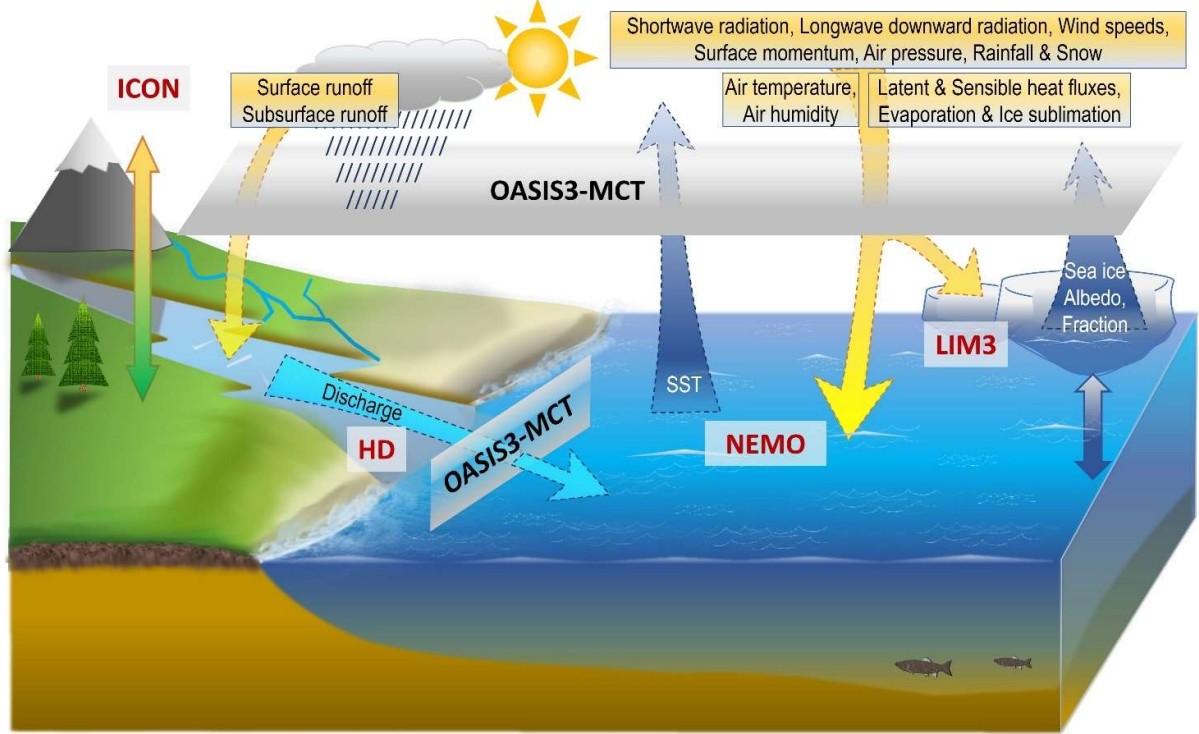

*Figure 1: Model components of GCOAST-AHOI and variables exchanged via the OASIS3-MCT coupler. Two solid arrows display the communication between atmosphere-land (yellow-green arrow) and ocean-sea ice (gray-blue arrow), which is done via subroutines inside ICON-CLM and NEMO, respectively. Dotted arrows show the transfer between components via the OASIS interface. Yellow arrows present atmospheric transfer to ocean-sea ice and river runoff. The cyan arrow shows the discharge from the river to the ocean. Blue arrows demonstrate the transfer of sea surface temperature (SST) from the ocean as well as the sea ice albedo and sea ice fraction to the atmosphere.*

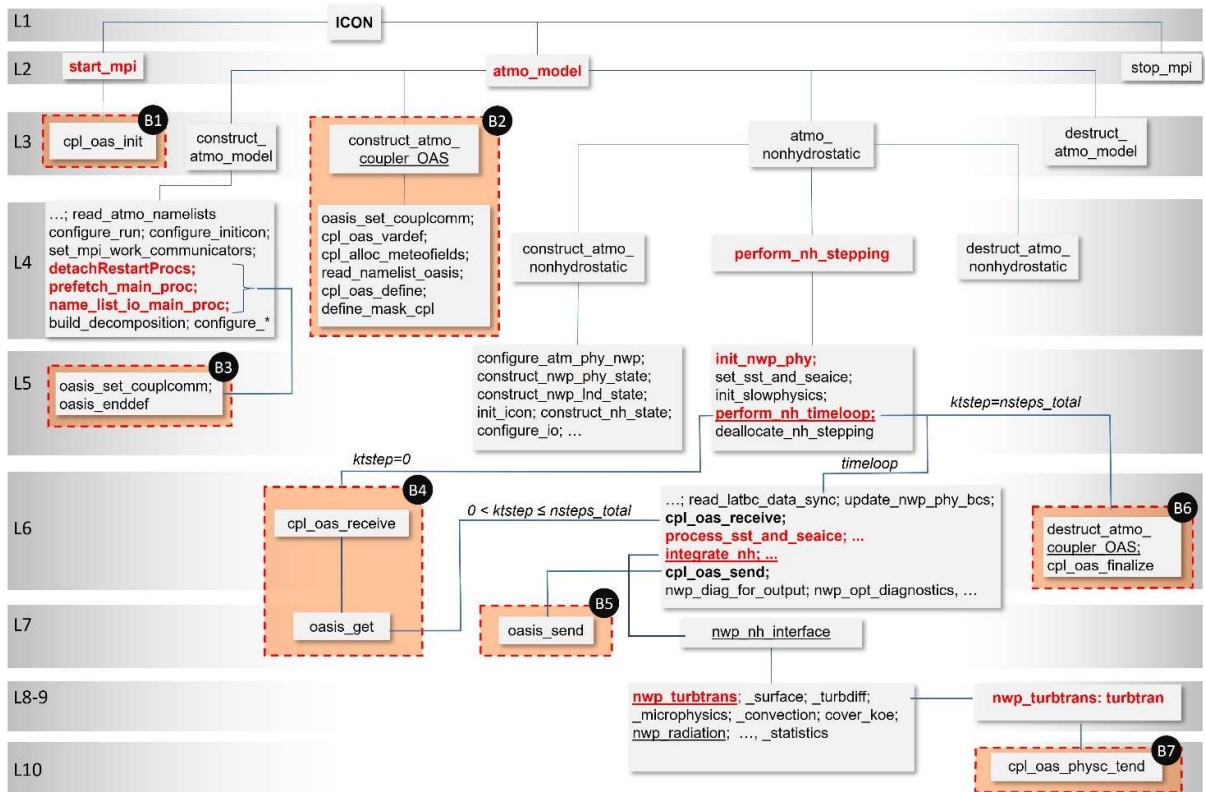


*Figure 2: Flowchart of ICON-NWP/ICON-CLM with the OASIS3-MCT coupling interface OMCI. The running sequence is from top to bottom, and from left to right. "L1" indicates the Level 1 – main program ICON, etc. At the levels 2 to 6, 8 and 9, subroutines (in red text) of ICON are modified by the coupling. At the levels 3 to 7 and 10, subroutines added for OMCI are shown in orange boxes (B1-B7).*

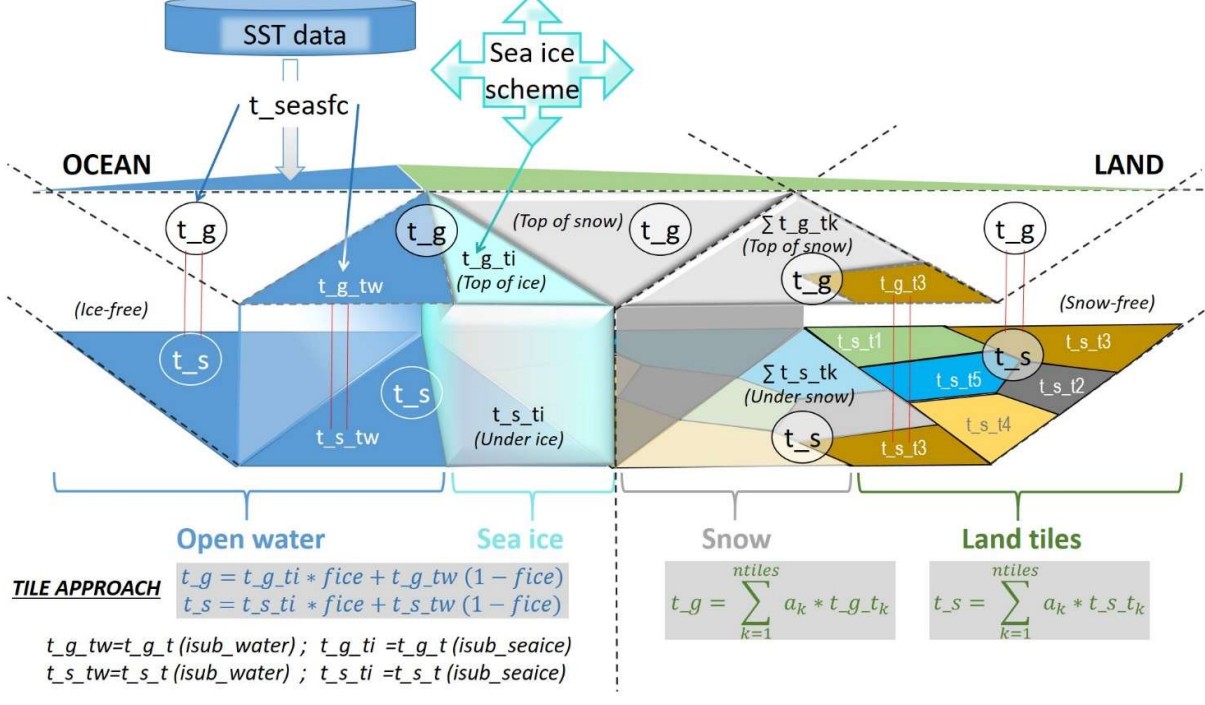


*Figure 3: Surface temperature exchange between atmosphere and ocean/land in ICON and GCOAST-AHOI.*

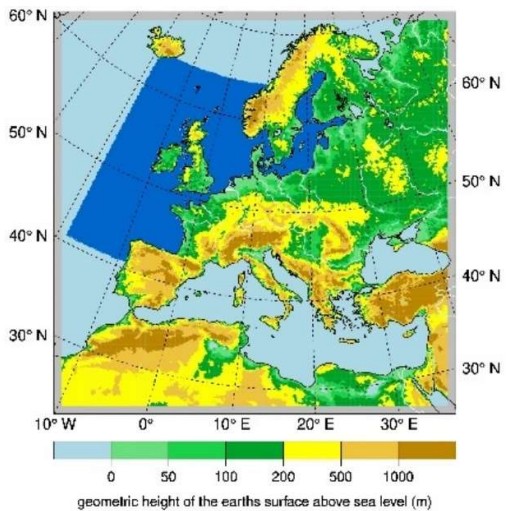


*Figure 4: Integration domains of ICON and HD (EURO-CORDEX) and of NEMO-LIM3 (dark blue).*


a) ICPL266

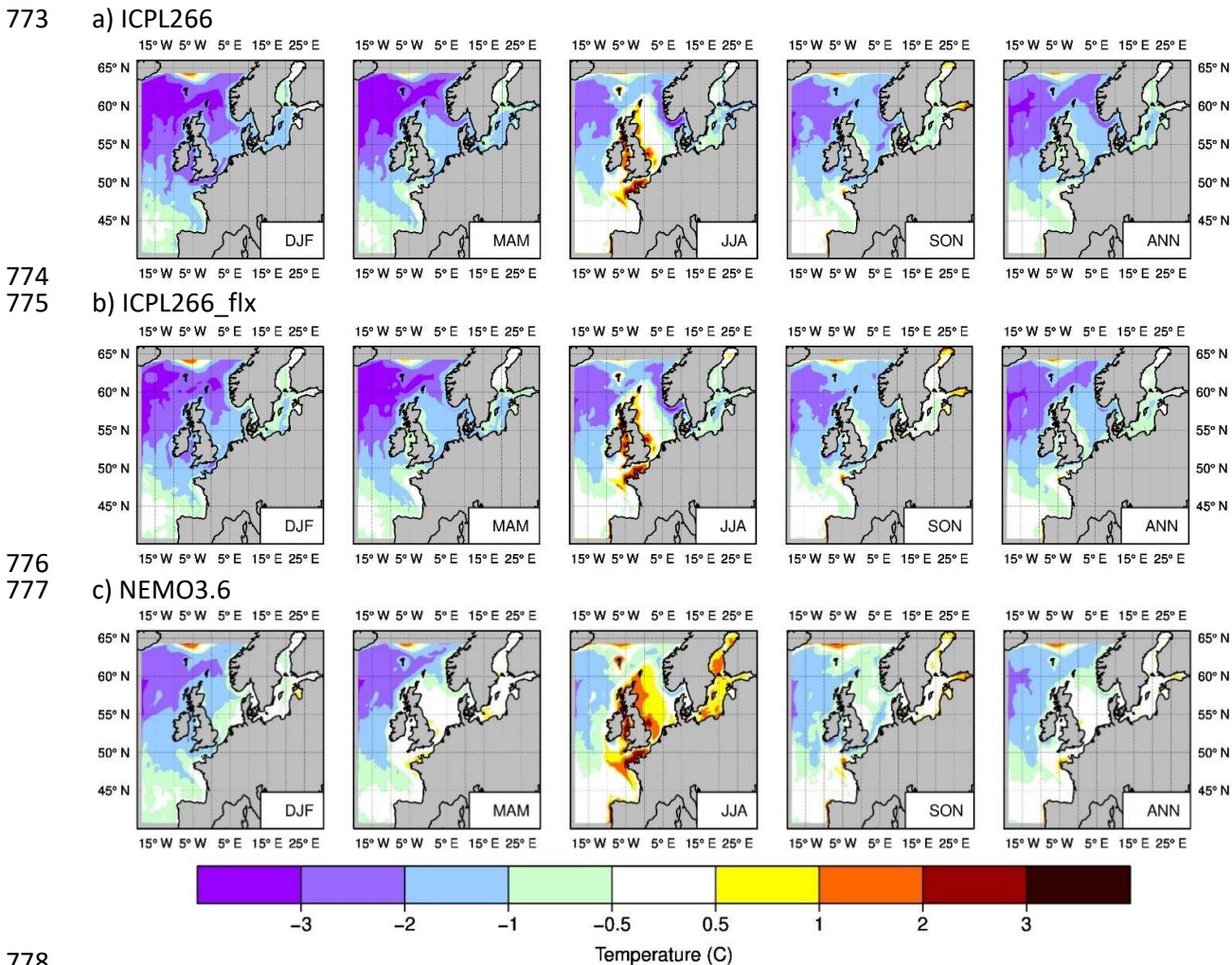

b) ICPL266_flx
c) NEMO3.6

*Figure 5: Seasonal (DJF, MAM, JJA, SON) and annual (ANN) mean of sea surface temperature (K) bias of a) ICPL266, b) ICPL266_flx and c) NEMO3.6 with respect to the OSTIA data for the period of 2010-2018 over the GCOAST domain.*


a) ICON266

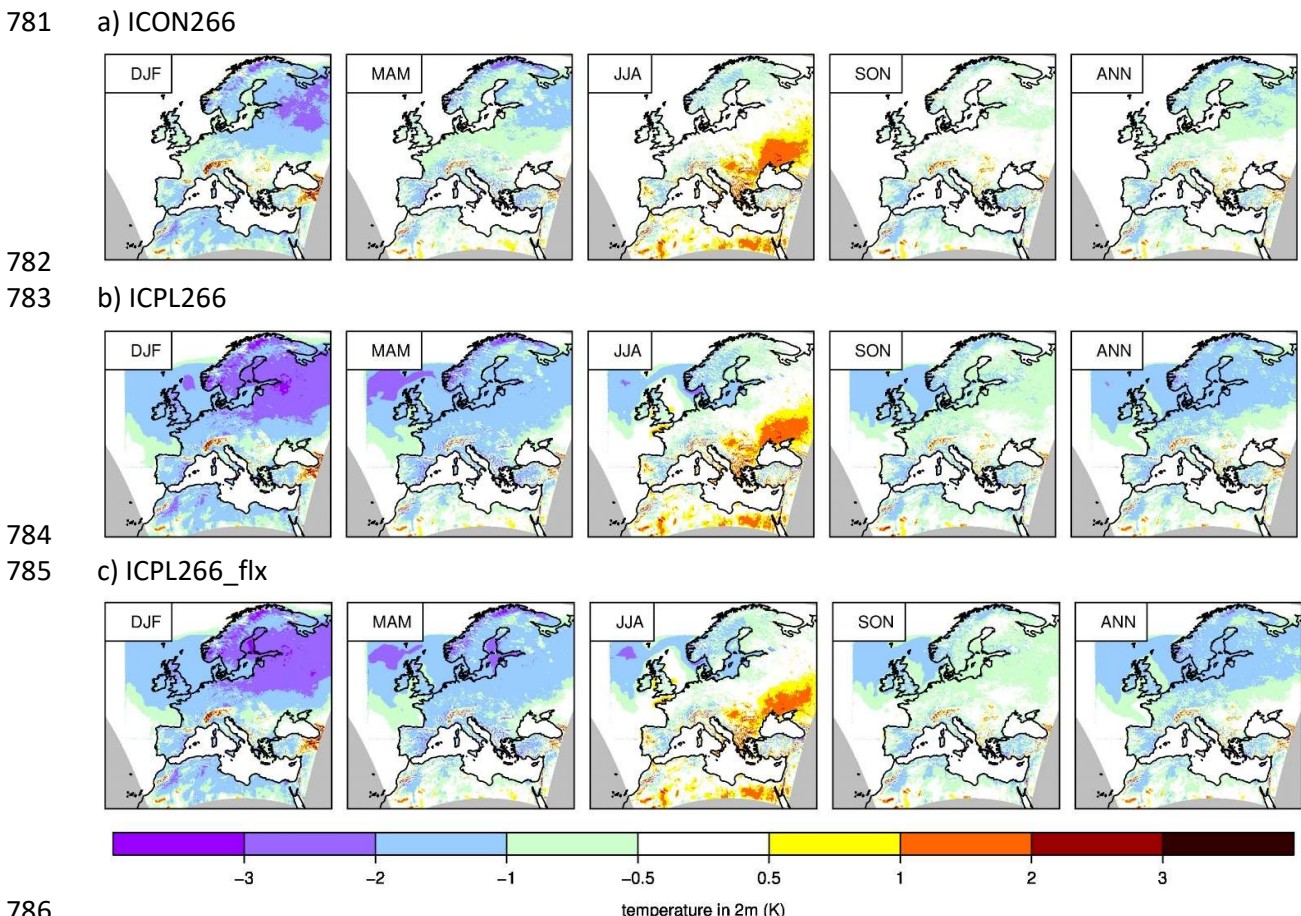

b) ICPL266
c) ICPL266_flx

*Figure 6: Seasonal (DJF, MAM, JJA, SON) and annual (ANN) mean of 2m air temperature (K) difference of a) ICON266, b) ICPL266 and c) ICPL266_flx to the ERA5 reanalysis data for the period of 2010-2018.*

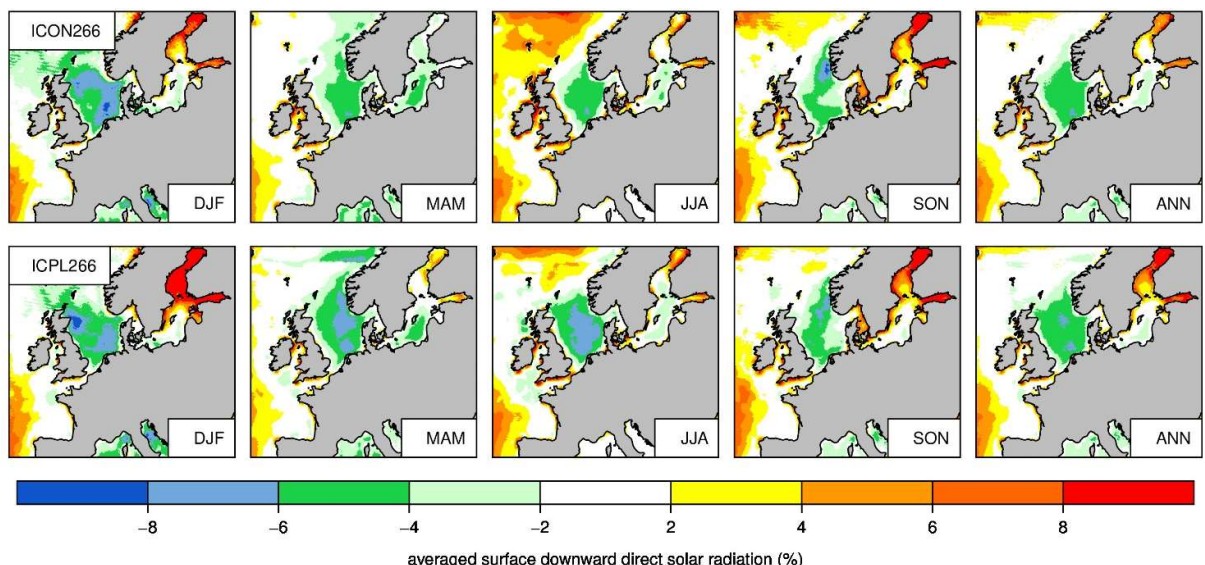

*Figure 7: Seasonal (DJF, MAM, JJA, SON) and annual (ANN) mean of shortwave downward radiation bias (%) of ICON266 (top) and ICPL266 (bottom) compared to the ERA5 data for the period of 2010-2018 over the GCOAST domain.*

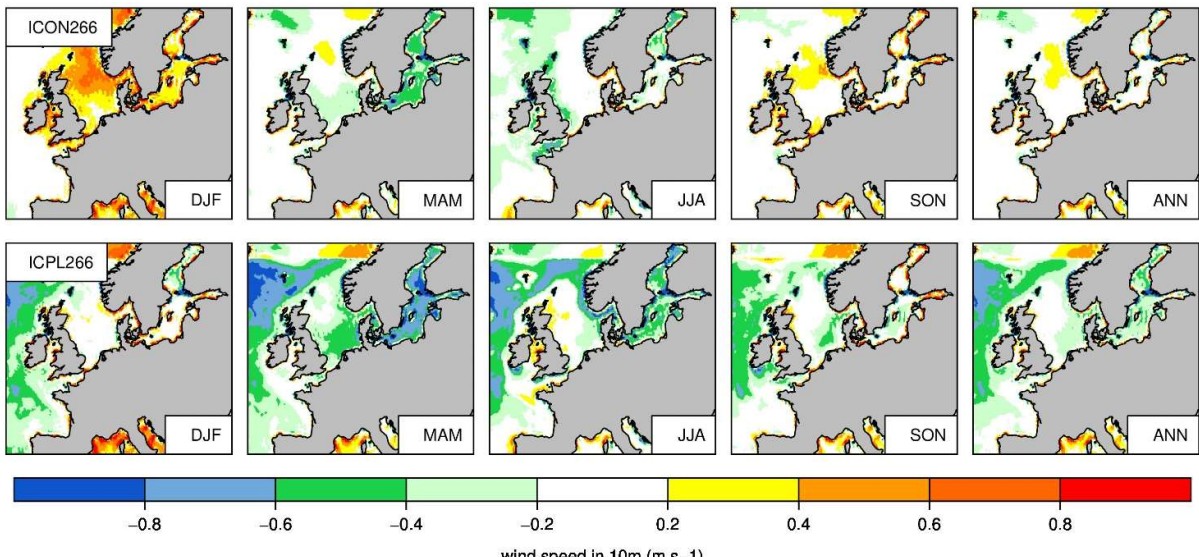

*Figure 8: Seasonal (DJF, MAM, JJA, SON) and annual (ANN) mean 10-M wind speed bias (m s $^{-1}$) of*
*ICON266 (top) and ICPL266 (bottom) compared to the ERA5 data for the period of 2010-2018 over*
*the GCOAST domain.*

a) ICON266

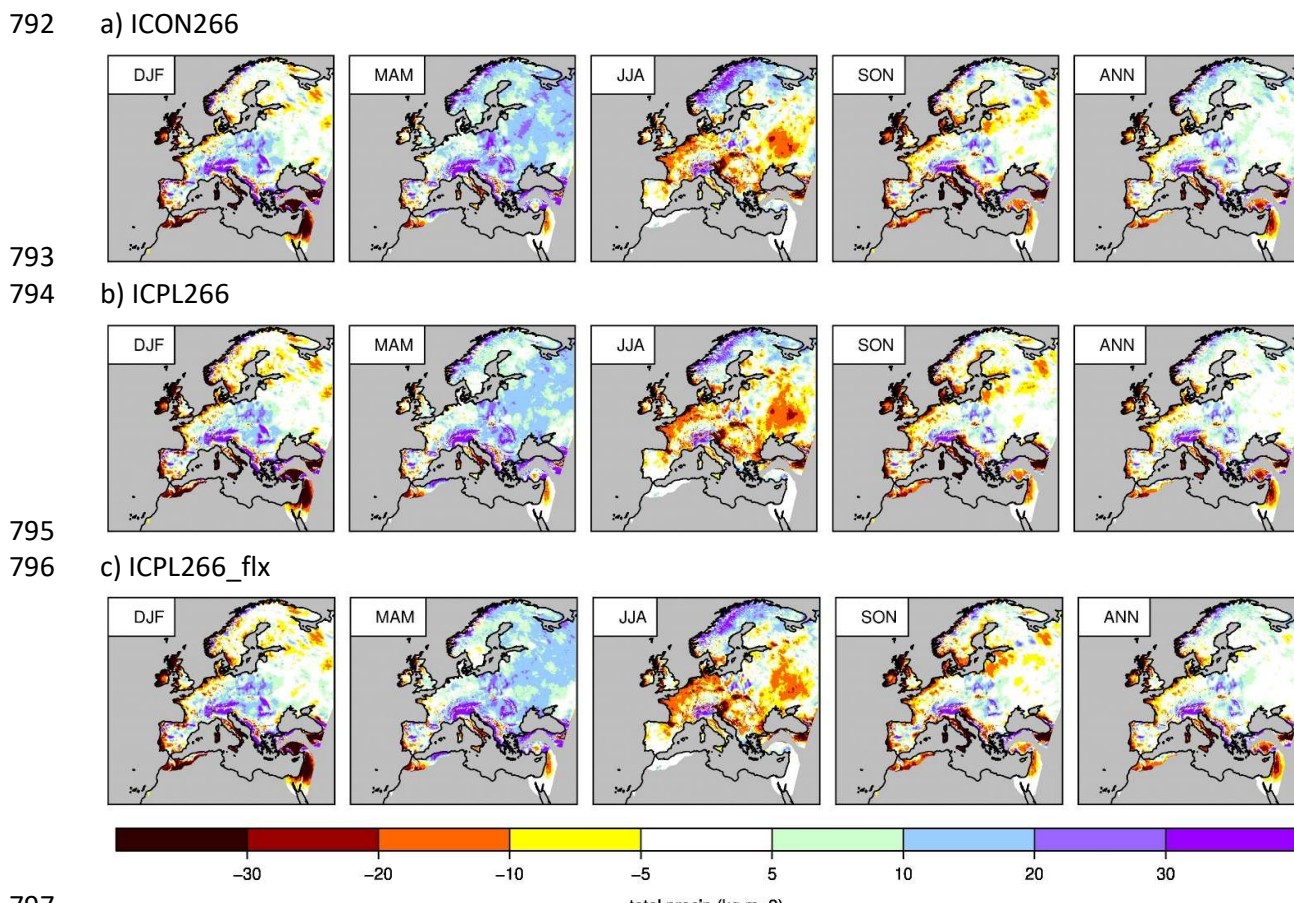

b) ICPL266
c) ICPL266_flx

*Figure 9: Seasonal (DJF, MAM, JJA, SON) and annual (ANN) difference of precipitation (mm month⁻¹) for a) ICON266, b) ICPL266 and c) ICPL266_flx compared to the E-OBS data for the period of 2010-2018.*


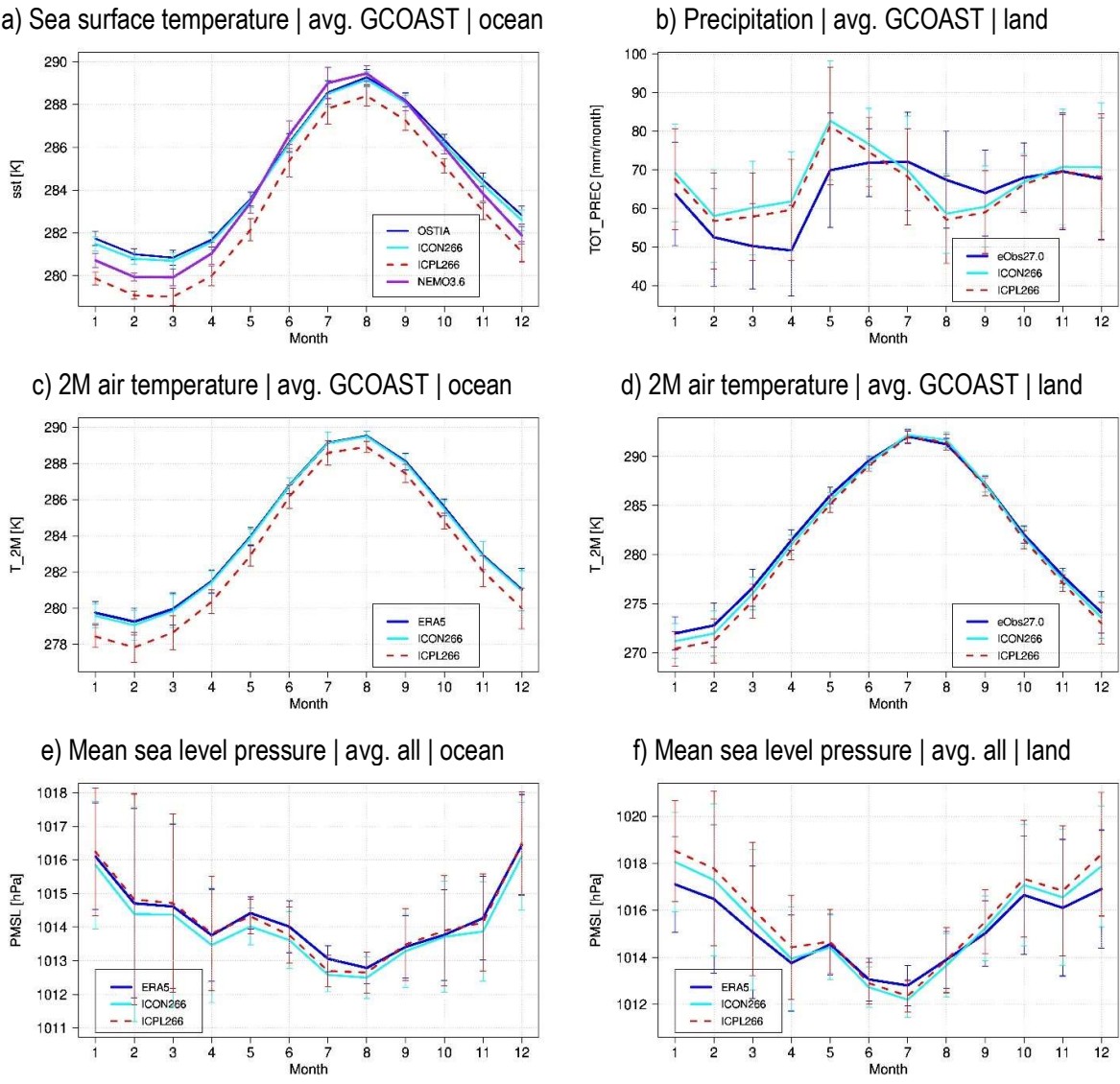

*Figure 10: Climatological monthly mean of T_S (K), T_2M (K), PMSL (hPa) and precipitation*
*(mm/month) of ICON266 (cyan solid line) and ICPL266 (red dashed line) compared to the OSTIA,*
*ERA5 and E-OBS data (blue solid line) for the period of 2010-2018. a) includes also the SST of NEMO*
*3.6 (purple solid line) averaged over the GCOAST domain. Values are averaged over the GCOAST*
*domain (avg. GCOAST) or the whole EURO-CORDEX domain (avg. all), over the ocean or land points*
*only. The vertical bars show the standard deviation of the area mean data.*

a) Sea ice fraction bias, ICPL266                    b) Sea ice fraction bias, ICPL266_flx

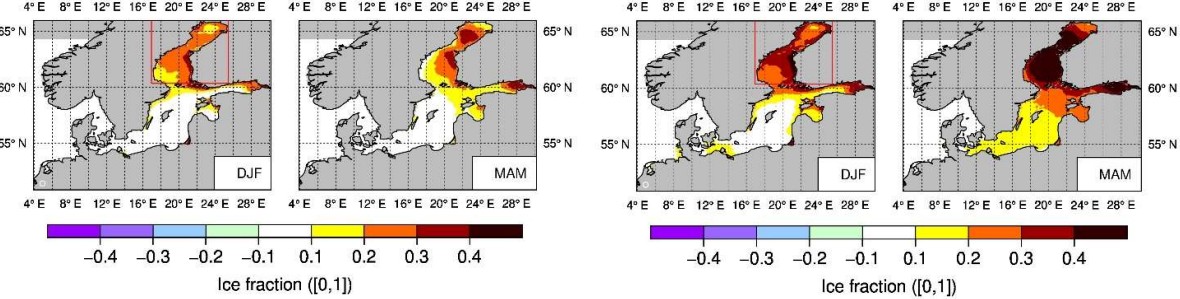

c) Sea ice fraction bias, NEMO3.6

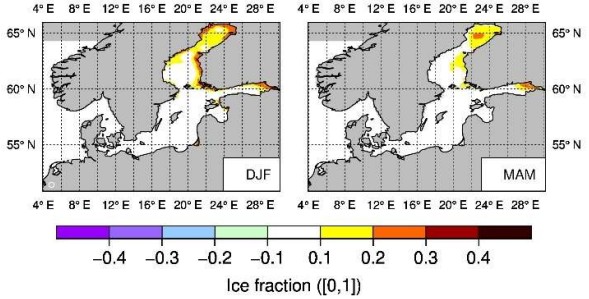

d) Sea ice fraction, Bothnian Bay & Sea

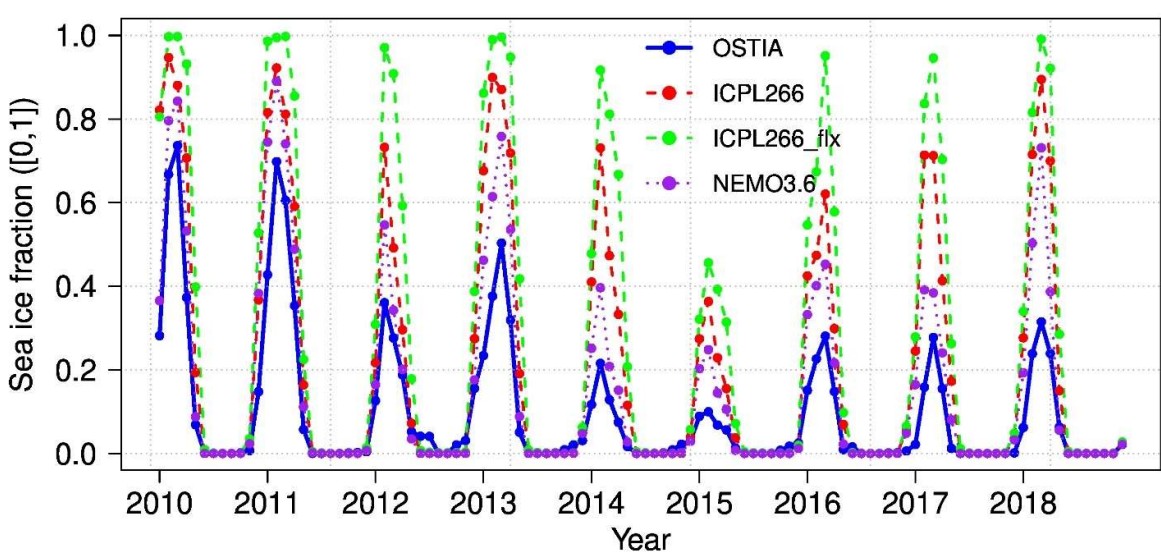

*Figure 11: Sea ice fraction bias of a) ICPL266, b) ICPL266_flx, c) NEMO3.6 compared to the OSTIA data in winter (DJF) and spring (MAM) averaged over the period of 2010-2018; d) Monthly mean of sea ice fraction averaged over the Bothnian Bay & Sea (red box in Fig. 11a) of OSTIA, ICPL266, ICPL266_flx and NEMO3.6 during 2010-2018.*


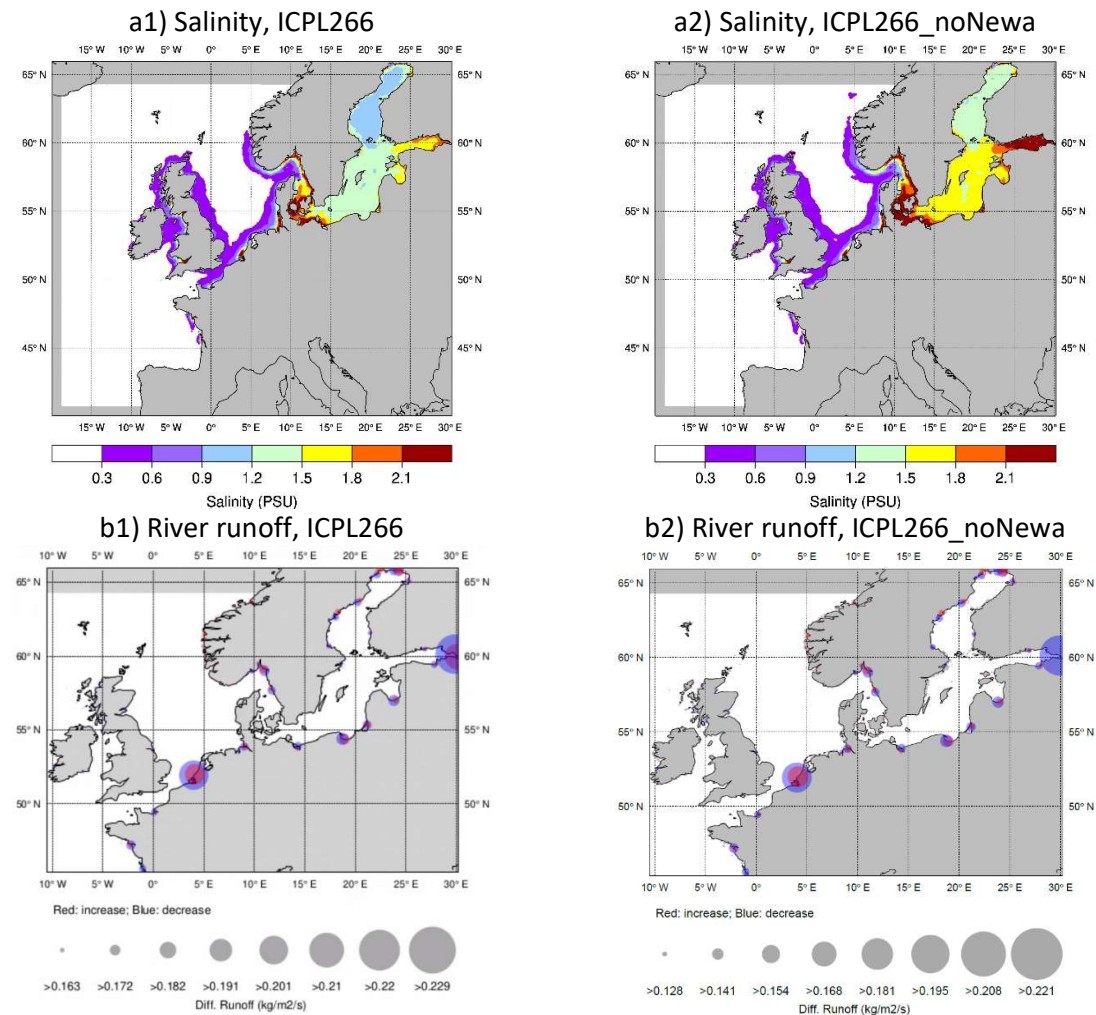

*Figure 12: a1) Salinity difference (PSU) and b1) river runoff difference (kg/m2/s) between ICPL266 and NEMO3.6 averaged over the period of 2010-2018. a2) and b2) salinity and river runoff difference of ICPL266_noNewa compared to NEMO3.6. In Fig. 12 b1 and b2, the size of the grey circles indicates the magnitude of the positive (red color) and negative (blue) differences.*
