# Peer review of "Coupling the regional climate model ICON-CLM v2.6.6 into the Earth system model GCOAST-AHOI v2.0 using OASIS3-MCT v4.0"

_EGUsphere, 2024_

## Referee Comment (RC1)

The manuscript on the coupling of the widely used atmosphere model ICON and the ocean model NEMO using OASYS3-MCT in a regional setup for the Baltic Sea, North Sea and parts of the Northeast Atlantic by Ho-Hagemann and co-workers is well-written and interesting. Although parts of the method section are very technical (code specific), a non-specialized reader can easily follow the presented text and analysis. The results, a comparison of two 10-year simulations driven by ERA5 at the lateral boundaries (coupled and uncoupled) with various datasets and ERA5 itself, are acceptable and in the quality range of other regional climate models. The authors analyzed sea surface temperature (SST), 2m air temperature, short-wave downward radiation, wind speed, precipitation and mean sea level pressure. I suggest to publish this article in GMD after minor to moderate revision.

General comments:

1) The analysis is too simple because you show only SST from the ocean and no results from the land model. As the freshwater input from land is very important for coastal seas like the Baltic Sea and North Sea, you should evaluate the results of the simulated river discharge from the HD model into the coastal seas. I understood, that the river discharge in the presented coupled simulation is from the HD model and not from observations. If you do not reproduce the integrated freshwater input (precipitation minus evaporation over land and over the ocean), the added value of the regional coupled model for the coastal ocean is rather limited. If the results are not sufficiently good, you may refer to further model development. However, please show these results because they would be interesting for the model performance in the ocean, e.g. for salinity.
2) I would like to see also an analysis of sea ice in the Baltic Sea, because the sea ice cover is very sensitive to biases in the surface heat and momentum fluxes over the coupling domain.
3) Furthermore, the integrated heat content changes could be evaluated by temperature profiles. Please use temperature and salinity data from the national monitoring programs for the North Sea and Baltic Sea to evaluate the ocean model performance and surface fluxes in the coupled model.
4) It would be good to present a list of scientific and technical arguments for your modeling strategy using OASYS3-MCT instead of YAC. For me the advantages of OASYS3-MCT compared to YAC are not completely clear.

Minor comments:

Line 303 Northern Africa

Figure 11: "Climatological monthly mean …" instead of "Annual variability …". Please add error bars or shaded areas to your curves denoting the interannual variability. This information is important to judge about the model-data differences.

---

## Author Response (AR1)

**Coupling the regional climate model ICON-CLM v2.6.6 into the Earth system model GCOAST-AHOI v2.0 using OASIS3-MCT v4.0**

We express our thanks to the two reviewers and Moritz Hanke for their comments which help us a lot to improve our manuscript.

Our answers are given in blue text below each comment.

**Reply to Reviewer#1 comments:**
**>> General comments:**
>> 1) The analysis is too simple because you show only SST from the ocean and no results from the land model. As the freshwater input from land is very important for coastal seas like the Baltic Sea and North Sea, you should evaluate the results of the simulated river discharge from the HD model into the coastal seas. I understood, that the river discharge in the presented coupled simulation is from the HD model and not from observations. If you do not reproduce the integrated freshwater input (precipitation minus evaporation over land and over the ocean), the added value of the regional coupled model for the coastal ocean is rather limited. If the results are not sufficiently good, you may refer to further model development. However, please show these results because they would be interesting for the model performance in the ocean, e.g. for salinity.

We add a new table (Table 3) of discharge and a new figure 12 of salinity and river runoff together with a new sub-section 5.5 (**L.378-449**, text is highlighted in red in the revised manuscript) to analyse the simulated salinity and river runoff.

**L.378-449:**

[revised manuscript text omitted]

>> 2) I would like to see also an analysis of sea ice in the Baltic Sea, because the sea ice cover is very sensitive to biases in the surface heat and momentum fluxes over the coupling domain.

We add a new figure 11 and a new sub-section 5.4 (**L.355-377**, text is highlighted in red in the revised manuscript) to analyse the simulated sea ice.

**L.355-377:**

"**5.4    Sea ice**

The sea ice fraction bias of ICPL266 is about 0.2-0.3 over the Bothnian Bay and Sea in winter and spring (Fig. 11a), while ICPL266_flx has a larger bias of about 0.3-0.5 (Fig. 11b) and the ERA5-forced NEMO3.6 has a relatively small positive ice fraction bias there (Fig. 11c). The monthly mean sea ice fraction averaged over the Bothnian Bay and Sea from ICPL266 and NEMO3.6 compared to the OSTIA data is shown in Fig. 11d, where the sea ice temporal variation is quite well captured by the two models, with a high peak in spring 2010 and a low peak in spring 2015. However, all three simulations overestimate the sea ice fraction of OSTIA, with the two coupled simulations showing a larger positive bias also in the time series. While ICPL266 has a winter SWDN about 8 % larger than ERA5 (Fig. 7b), the incoming shortwave radiation is relatively small over the high latitudes in winter. Therefore, we don't expect the positive SWDN bias to be the main reason for the overestimation of sea ice. However, the LWDN of ICPL266 is about 10 W/m2 lower than ERA5 (the forcing for NEMO3.6) in winter over the North and Baltic Seas (Figs. S6b, S7b), and T_2M is about 3 °C lower than ERA5 over the Scandinavian region surrounding the Bothnian Bay and Sea (Fig. 6b). The cold T_2M bias and negative LWDN bias of ICPL266 may explain its positive sea ice fraction bias. The cold air temperature above the sea ice surface often produces more sea ice in winter and spring, especially over an area with water of low salinity, such as the Bothnian Bay and Sea. Figure 6c shows the larger T_2M cold bias of ICPL266_flx over the Baltic Sea in spring about 1oC more than ICPL266 (Fig. 6b) which is consistent with the larger sea ice fraction bias of ICPL266_flx (Fig. 11b) compared with that of ICPL266 (Fig. 11a).

Another factor that could contribute to an increase in sea ice cover in spring would be an increase in river runoff, which would result in less salty sea water and therefore more sea ice. These two variables are analyzed in the next section."

>> 3) Furthermore, the integrated heat content changes could be evaluated by temperature profiles. Please use temperature and salinity data from the national monitoring programs for the North Sea and Baltic Sea to evaluate the ocean model performance and surface fluxes in the coupled model.

We add four new figures S6, S7, S8, S9 about the surface fluxes in the Supplementary and an analysis in section 5.2 (**L.279-310**, text is highlighted in red in the revised manuscript).

**L.279-310:**

".. Figures S6a, b and Figures S7 a, b show the seasonal SWDN and LWDN of ICPL266 and NEMO3.6 averaged over the North Sea and Baltic Sea for the period of 2010-2018. Note that we don't show the result of ICPL266_flx in Figs. S6 and S7 because there is no output of LWDN in ICPL266_flx due to the setup of the CPL_flx coupling method. Over the North Sea, the SWDN of ICPL266 is smaller than the ERA5 used for NEMO3.6 in spring and summer (Fig. S6a), which mainly leads to the cold SST bias of ICPL266 (Fig. 5a). Therefore, we plan to increase the SWDN of ICON by about 10 % before sending it to NEMO. However, the cold SST bias over the Baltic Sea does not seem to be directly related to the SWDN as there is no clear SWDN difference between ICPL266 and NEMO3.6 in summer or in any other season (Fig. S6a). The LWDN of ICPL266 is similar to NEMO3.6 in summer but slightly smaller in the other three seasons, both over the North Sea and the Baltic Sea. Increasing the LWDN in ICON-CLM by about 5-10 W/m2 before sending it to NEMO should be tested to reduce the SST bias. Note that the seasonal cycle of LWDN is more pronounced over the North Sea than over the Baltic Sea.

We also compare the turbulent heat flux (i.e. the sum of SH and LH) of NEMO3.6 and the flux of NEMO in ICPL266, averaged over the North Sea and the Baltic Sea (Fig. S6c and Fig. S7c), and the net downward heat flux, which is the sum of SWDN, LWDN, SH and LH (Fig. S6d and Fig. S7d). We only consider the turbulent flux because NEMO doesn't write out SH and LH separately, but only the SWDN, LWDN and the net downward heat flux. Note that the turbulent flux from NEMO3.6 is calculated using the "CORE bulk formulae", and due to the CPLmix coupling method used, the turbulent flux in NEMO from ICPL266 is the average of the flux from ICON-CLM and the one calculated inside NEMO (see section 3.2). The results are similar for both seas. The turbulent flux and the net downward flux of the two experiments are quite similar with the largest differences in winter (DJF) and summer (JJA).

Using ERA5 as a reference, the SH and LH biases of ICON266 and ICPL266 are shown in Fig. S8 and Fig. S9. Over land, the bias of ICON266 is very similar to that of ICPL266. However, over the ocean, the bias of ICPL266 is generally more positive (i.e. the fluxes are less negative) than that of ICON266 with the largest bias over the North Atlantic. Smaller heat fluxes are consistent with lower SSTs in ICPL266, as lower SSTs lead to larger stability and less vertical mixing. Over the North and Baltic Seas, SH and LH of ICPL266 are quite close to ERA5. Despite of the SST forcing from ERA5, ICON266 has a negative SH bias of about -5 to -15 W/m2 over the North and Baltic Seas, especially in winter. This suggests a future analysis of the difference in air temperature at the lowest level of ICON-CLM and ERA5."

The temperature profile analysis will be done in a future study when a new simulation with the optimal settings of ICON-CLM will be conducted for a longer time period.

>> 4) It would be good to present a list of scientific and technical arguments for your modeling strategy using OASIS3-MCT instead of YAC. For me the advantages of OASIS3-MCT compared to YAC are not completely clear.

There is no scientific argument for preferring one external coupler over the other in ICON-CLM. Both couplers, OASIS3-MCT and YAC, provide the tools for coupling climate component models of the Earth system.

However, there are several technical and practical arguments for using OASIS3-MCT. Hence, we rewrite the paragraphs:

**L.61-74:**

"For the coupling of ICON-CLM as an atmosphere component into GCOAST-AHOI, which includes HD and the ocean model NEMO (Nucleus for European Modelling of the Ocean, Madec et al. 2017), representing the ocean and sea ice components, basically, there were two feasible options: either to implement a YAC interface in NEMO and HD, or to implement an OASIS interface in ICON-CLM. For the option one, the YAC coupling interface was added into the HD source code by M. Hanke (DKRZ) (see Hagemann et al. 2023), but YAC has not yet been available in the NEMO source code. To our current knowledge, there is no RESM with NEMO using the YAC coupler. The NEMO model is already linked to the OASIS coupler, which has been used to couple NEMO with many other model components. Implementing the YAC interface in NEMO would require a larger effort, as the NEMO source code is much more complicated than the HD code. In addition, although the NEMO source code is freely available, we are ordinary users in the NEMO community, not members of the model development team. Therefore, implementing and especially maintaining the YAC interfaces in NEMO is a big challenge."

**L.490-493:**

"The new OASIS3-MCT coupling interface OMCI implemented in the ICON-CLM model adds a possibility to couple ICON-CLM with an external ocean model and an external hydrological discharge model, not only with NEMO and HD, using OASIS3-MCT instead of YAC."

And modify the abstract:

"The new OASIS3-MCT coupling interface OMCI implemented in the ICON-CLM model adds a possibility to couple ICON-CLM with an external ocean model and an external hydrological discharge model using OASIS3-MCT instead of the YAC coupler. Using OMCI, it is also possible to set up a RESM with ICON-CLM and other ocean and hydrology models possessing the OASIS3-MCT interface for other regions, such as the Mediterranean Sea."

**>> Minor comments:**
>> Line 303 Northern Africa
Corrected. Thank you!

>> Figure 11: "Climatological monthly mean …" instead of "Annual variability …". Please add error bars or shaded areas to your curves denoting the interannual variability. This information is important to judge about the model-data differences.

Done. Thank you!
* * *
**Reply to Reviewer#2 comments:**
**>> General comments:**

>> From line 190 to 198, the authors describe the three coupling techniques and, in the end, they chose the mixing strategy, where NEMO averages the LH and SH send by ICON (fluxes calculated based on tile approach) and LH and SH calculated using bulk formulae. As the authors cite, this approach was also employed by Ho-Hagemann et al. (2020). In Ho-Hagemann et al. (2020), about the coupling of CCLM with NEMO, it was said that the default coupling strategy based on flux exchange from the atmosphere lead to large biases on SST, and that's why they preferred to use the average of fluxes from both models based on different turbulence parameterizations. Even if could be true that the analysis of the best coupling strategy could go beyond the scope of this paper and could be matter of further studies, it could be useful to see an evaluation analysis of the turbulent fluxes component at surface against observations or reanalysis.

We add four new figures S6, S7, S8, S9 about the surface fluxes in the Supplementary and an analysis in section 5.2 (**L.279-310**, text is highlighted in red in the revised manuscript). Please read the text above at the reply to comment 3 of the Reviewer#1.

A new experiment ICPL266_flx using the coupling method CPL_flx is conducted and analyzed in section 5.

We also add some sentences about the different results of coupling COSMO-CLM and ICON-CLM to NEMO:

**L237-242:**

> "Ho-Hagemann et al. (2020) noted that using CPL_flx when coupling COSMO-CLM with NEMO leads to larger biases in the SST than using CPL_mix. This is not the case here when coupling ICON-CLM. A possible reason for this is that due to the tile approach (cf. Sect. 3.2), the fluxes from ICON-CLM to NEMO are sent separately over water and sea ice, while COSMO-CLM v5.0 doesn't have the tile approach, therefore, the fluxes in each ocean grid box sent from the atmosphere to the ocean are the mixed fluxes of water and sea ice."

>> The authors should also include an evaluation analysis for the hydrological discharge component, this is extremely important to close the balance of the freshwater. If the results are unsatisfactory, they should be highlighted as current model limitation and, maybe, as further model improvement in future works.

We add a new table (Table 3) of discharge and a new figure 12 of salinity and river runoff together with a new sub-section 5.5 (**L.378-449**, text is highlighted in red in the revised manuscript) to analyse the simulated salinity and river runoff. Please read the text above at the reply to comment 1 of the Reviewer#1.

**>> Minor comments:**

>> I found the interface structure part (lines 104-180) and the profiling part of the model with the LUCIA tool (lines 246-279) too technical and verbose. These parts could be summarized in the main core of the paper, while all the technicalities, which are extremely useful for the developers and users of both community models and for the reproducibility of the study, could go in the supplement material.

Thank you for this suggestion. We agree with you and move those parts to the Supplementary. The Table 2 is moved to be the Table S3 in the Supplementary. We include a new Table 2 in the main text to list the model experiments.

>> Some of the figure results regarding the NEMO standalone biases are just described and bear the wording "not shown", maybe also these figures could go in the supplement material? Even if these figures could not give any added value for the results, they could help to visualize the NEMO standalone biases.

We add some figures of stand-alone NEMO in Fig. 5c, Fig. 11, Fig. 12, Fig. S6 and Fig. S7 and the analysis in the section 5.

>> From line 46 to 51, the authors describe why they chose to implement OASIS in ICON and not YAC in NEMO. Could they elaborate on why the implementation of YAC in NEMO is a major challenge than the implementation of OASIS in ICON? I think that, besides the different coupler architectures and implementation issues, the main driver of this decision has been lack of a regional version of the ICON-O component, currently under development as explained in the results.

The reviewer is right. We rewrite some paragraphs and the abstract to provide the technical arguments for using OASIS3-MCT. Please read the text above at the reply to comment 4 of the Reviewer#1.
* * *
**Reply to community comment:**

>> The paper gives the impression that with the introduction of OMCI, ICON is able to be coupled to external (non-ICON) components (see abstract or lines 407-409). However, in its current state, ICON is already able to do that. OMCI adds the possibility to do this using OASIS3-MCT instead of YAC.
We rewrite the sentence **L.490-493:**

"The new OASIS3-MCT coupling interface OMCI implemented in the ICON-CLM model adds a possibility to couple ICON-CLM with an external ocean model and an external hydrological discharge model, not only with NEMO and HD, using OASIS3-MCT instead of YAC."

And modify the abstract:

"The new OASIS3-MCT coupling interface OMCI implemented in the ICON-CLM model adds a possibility to couple ICON-CLM with an external ocean model and an external hydrological discharge model using OASIS3-MCT instead of the YAC coupler. Using OMCI, it is also possible to set up a RESM with ICON-CLM and other ocean and

hydrology models possessing the OASIS3-MCT interface for other regions, such as the Mediterranean Sea."

>> ICON consists of multiple components (e.g. atmosphere, ocean, and wave). The paper sometimes does not to distinguish between ICON and its components. It is also not clearly stated that OMCI supports the coupling of a single ICON component (ICON atmosphere) to an external one. Coupling between ICON components, or the coupling of multiple ICON components with an external one is not supported and potentially impossible due to how the initial communicator splitting is implemented.
We rewrite the introduction section to distinguish between the ICON components. In addition, "ICON-CLM" in the title refers to the atmospheric component of ICON.

**Specific comments:**
>> Line 49-50: Can you elaborate why it is a major challenge to introduce a new coupling interface to NEMO and why this not the case for ICON?
We rewrite the paragraph as follows:
**L.61-74:**
> "For the coupling of ICON-CLM as an atmosphere component into GCOAST-AHOI, which includes HD and the ocean model NEMO (Nucleus for European Modelling of the Ocean, Madec et al. 2017), representing the ocean and sea ice components, basically, there were two feasible options: either to implement a YAC interface in NEMO and HD, or to implement an OASIS interface in ICON-CLM. For the option one, the YAC coupling interface was added into the HD source code by M. Hanke (DKRZ) (see Hagemann et al. 2023), but YAC has not yet been available in the NEMO source code. To our current knowledge, there is no RESM with NEMO using the YAC coupler. The NEMO model is already linked to the OASIS coupler, which has been used to couple NEMO with many other model components. Implementing the YAC interface in NEMO would require a larger effort, as the NEMO source code is much more complicated than the HD code. In addition, although the NEMO source code is freely available, we are ordinary users in the NEMO community, not members of the model development team. Therefore, implementing and especially maintaining the YAC interfaces in NEMO is a big challenge."

>> Line 52: Does the same argument for not having a need for a YAC interface in NEMO also applies to ICON?
We remove the sentence and rewrite the paragraph. Please read above.

>> Paragraph line 423-431: Since ComIn does not allow a plugin to change the communicators of ICON, the current OMCI approach cannot be simply implemented as a ComIn-Plugin.
Thanks a lot for your comment and the information. We rewrite this paragraph:
**L.506-514:**
> "Recently, the ICON Consortium has developed and released the Community Interface (ComIn) for the ICON model to allow ICON to be coupled with external model components. The main challenge for the external model component coupling is the initial splitting of MPI_COMM_WORLD, which is done in ICON by a grouping of the mpi communicators (MPI-handshake) (M. Hanke, pers. communication, 2024). There are about 40 ComIn entry points in the new release version of ICON. Using the ComIn entry points

does not require any additional patching of the ICON source code. A coupling interface to an external model such as OMCI must be moved into a ComIn-plugin to connect to the entry points in the ICON source code. In addition, the communicator splitting using the MPI-handshake algorithm must be implemented in the NEMO and HD source code."